# Hypertrophic chondrocytes serve as a reservoir for marrow-associated skeletal stem and progenitor cells, osteoblasts, and adipocytes during skeletal development

**Jason T Long[1,2], Abigail Leinroth[1,2], Yihan Liao[2,3], Yinshi Ren[2], Anthony J Mirando[2], Tuyet Nguyen[4], Wendi Guo[2,3], Deepika Sharma[2], Douglas Rouse[5], Colleen Wu[1,2,3], Kathryn Song Eng Cheah[6], Courtney M Karner[1,2], Matthew J Hilton[1,2]***

[1]Department of Cell Biology, Duke University School of Medicine, Durham, United States; [2]Department of Orthopaedic Surgery, Duke University School of Medicine, Durham, United States; [3]Department of Pharmacology and Cancer Biology, Duke University School of Medicine, Durham, United States; [4]Program of Developmental and Stem Cell Biology, Duke University School of Medicine, Durham, United States; [5]Division of Laboratory Animal Resources, Duke University School of Medicine, Durham, United States; [6]School of Biomedical Sciences, University of Hong Kong, Hong Kong, Hong Kong

**\*For correspondence:**
matthew.hilton@duke.edu

**Abstract** Hypertrophic chondrocytes give rise to osteoblasts during skeletal development; however, the process by which these non-mitotic cells make this transition is not well understood. Prior studies have also suggested that skeletal stem and progenitor cells (SSPCs) localize to the surrounding periosteum and serve as a major source of marrow-associated SSPCs, osteoblasts, osteocytes, and adipocytes during skeletal development. To further understand the cell transition process by which hypertrophic chondrocytes contribute to osteoblasts or other marrow associated cells, we utilized inducible and constitutive hypertrophic chondrocyte lineage tracing and reporter mouse models (*Col10a1CreERT2; Rosa26*^fs-tdTomato^ and *Col10a1Cre; Rosa26*^fs-tdTomato^) in combination with a *PDGFRa*^H2B-GFP^ transgenic line, single-cell RNA-sequencing, bulk RNA-sequencing, immunofluorescence staining, and cell transplantation assays. Our data demonstrate that hypertrophic chondrocytes undergo a process of dedifferentiation to generate marrow-associated SSPCs that serve as a primary source of osteoblasts during skeletal development. These hypertrophic chondrocyte-derived SSPCs commit to a CXCL12-abundant reticular (CAR) cell phenotype during skeletal development and demonstrate unique abilities to recruit vasculature and promote bone marrow establishment, while also contributing to the adipogenic lineage.

## Editor's evaluation

The work reveals that a subpopulation of hypertrophic chondrocytes can form SSPCs that give rise to osteoblasts and adipocytes during skeletal development, providing new evidence to the current concept of chondrocyte-osteoprogenitors-osteoblast trans-differentiation.

## Introduction

Most skeletal elements arise from a cartilage template via the process of endochondral ossification (*Long and Ornitz, 2013*; *Olsen et al., 2000*; *Ono et al., 2019*). During skeletal development, mesenchymal stem and progenitor cells condense and differentiate to form chondrocytes or cartilage cells (*Hirao et al., 2006*; *Maes et al., 2012*; *Olsen et al., 2000*; *Spencer et al., 2014*). These chondrocytes undergo rapid proliferation while depositing an extracellular matrix (ECM) rich in type II collagen (COL2A1), forming the initial immature cartilage rudiments (*Ono et al., 2019*). As cartilage growth proceeds, the chondrocytes localized nearest the center of each rudiment exit the cell cycle and undergo a process of maturation and hypertrophic differentiation (*Capasso et al., 1984*; *Kong et al., 1993*; *Kwan et al., 1997*; *Long and Ornitz, 2013*; *Olsen et al., 2000*; *Ono et al., 2019*; *Reginato et al., 1986*; *Schmid and Conrad, 1982*). Hypertrophic chondrocytes, which are identified by their specific expression of type X collagen (COL10A1), are ultimately responsible for secreting additional proteins such as Vascular Endothelial Growth Factor alpha (VEGFα), Platelet Derived Growth Factor beta (PDGFβ), and Indian Hedgehog (IHH) among other factors to promote both vascular invasion and osteoblast differentiation of the surrounding perichondrial progenitors (*Capasso et al., 1984*; *Kwan et al., 1997*; *Olsen et al., 2000*; *Reginato et al., 1986*; *Schmid and Conrad, 1982*; *St-Jacques et al., 1999*; *Zelzer et al., 2004*). As vessels from the perichondrium invade the hypertrophic cartilage to establish the marrow cavity, newly formed osteoblasts and their progenitors are delivered by these vessels to the marrow space (*Maes et al., 2010*; *Olsen et al., 2000*). There, osteoblasts secrete a type I collagen (COL1A1)-rich matrix that ultimately mineralizes to replace the cartilage with bone, while the hypertrophic cartilage is remodeled and reduced with age (*Olsen et al., 2000*). Hypertrophic chondrocytes are essential for both establishment of the marrow cavity and bone formation; however, their precise fate and direct or indirect contribution to bone formation has been questioned (*Tsang et al., 2015*).

Recent genetic evidence suggests that hypertrophic chondrocytes undergo a process of transdifferentiation to directly generate osteoblasts and osteocytes that contribute to the formation of trabecular and endocortical bone (*Park et al., 2015*; *Yang et al., 2014b*; *Yang et al., 2014a*; *Zhou et al., 2014a*). To determine the fate of growth plate chondrocytes during endochondral bone formation, investigators have utilized mouse lines combining constitutive and inducible CRE recombinases under the control of the *Col2a1, Acan,* or *Col10a1* promotors/enhancers with a recombination sensitive reporter. Utilization of the constitutive and inducible *Col2a1Cre/Col2a1CreERT2* mouse lines have complicated our understanding of this lineage by the fact that in addition to their expression in immature chondrocytes, *Col2a1* and these genetic tools have been noted to be expressed within and/or label directly perichondrial and marrow associated cell populations known to give rise to osteoblast lineage cells (*Hilton et al., 2007*; *Long et al., 2001*). Some of the more convincing data generated utilizing a *Col2a1CreERT2* mouse line and recombination-sensitive reporter comes from *Yang et al., 2014a*, which established an approach for low-dose tamoxifen delivery to drive recombination in clones or clusters of immature chondrocytes that could be tracked over time. When the chase period was short (1–5 days post tamoxifen), reporter expression was largely restricted to immature chondrocyte clusters or extensions of these clusters into early hypertrophic chondrocytes. However, when the chase period was extended (10 days post tamoxifen), reporter expression could be observed throughout individual columns of proliferating to hypertrophic chondrocytes that also extended into *Col1a1*-expressing osteoblasts localized only to the regions of metaphyseal bone directly underneath the labeled chondrocytes; a result that is consistent with chondrocytes transitioning to an osteoblastic fate (*Yang et al., 2014a*). Similar pulse-chase experiments were performed utilizing the *AcanCre^ERT2*; *Rosa26^fs-tdTomato* reporter model at both embryonic and adult time points utilizing doses of tamoxifen that elicit high rates of recombination in chondrocytes. These data suggested that following short (1–3 days post tamoxifen) or long (2 weeks post tamoxifen) chase periods, non-chondrocytic cells associated with the primary spongiosa and trabecular bone were reporter positive with some cells labeling with the *Col1a1(2.3 kb)-eGFP*.(*Zhou et al., 2014b*) The most convincing data was generated utilizing a combination of constitutive and inducible *Col10a1Cre/Col10a1CreERT2* mouse lines and recombination-sensitive reporters; models that are much more restricted in their expression to hypertrophic chondrocytes of the growth plate (*Yang et al., 2014b*). This study, as well as others solely utilizing a constitutive *Col10a1Cre* model, determined that hypertrophic chondrocytes can relatively quickly give rise to osteoblastic cells expressing OSTERIX (OSX), COL1A1, and eventually

even OSTEOCALCIN (OCN) and/or SCLEROSTIN (SOST) expressing osteocytes within the trabecular and endocortical bone matrix (*Park et al., 2015*; *Yang et al., 2014b*; *Yang et al., 2014a*; *Zhou et al., 2014b*). Interestingly, *Yang et al., 2014a* utilized a *Col10a1Cre; Rosa26^{fs-eYFP}* reporter model and noted the presence of some YFP+, *Perilipin*-expressing adipocytes and YFP+, PDGFRβ+ pericytes within the bone marrow, suggesting the potential for alternate fates of hypertrophic chondrocytes beyond the osteoblast lineage. Utilizing similar techniques, *Tan et al., 2020* also demonstrated the potential for mature adipocytes to arise from *Col10a1*-expressing hypertrophic chondrocytes.

Since the fates of mature osteoblasts, adipocytes, and pericytes are unlikely to arise from independent hypertrophic chondrocyte transdifferentiation events, we reasoned that the hypertrophic chondrocytes may undergo a dedifferentiation process to generate skeletal stem and progenitor cell (SSPC) population(s) that localize to the bone marrow and maintain the capacity of redifferentiating into more mature cell fates. To address this issue, we utilized hypertrophic chondrocyte genetic reporter mouse models combined with single cell RNA-sequencing, immunofluorescent staining validation assays, fluorescent activated cell sorting (FACS)/flow cytometry, and bulk RNA-sequencing approaches of clonal cell populations. Our data indicate that hypertrophic chondrocytes indeed undergo a process of dedifferentiation to generate bone-marrow-associated SSPCs embryonically and postnatally. These SSPC likely serve as a primary source of osteogenic cells during skeletal development and also contribute to the adipogenic lineage, but not the pericyte lineage. Further, these studies indicate that the hypertrophic chondrocyte derived SSPCs are strikingly similar at the transcriptomic level to marrow-associated SSPCs derived from other sources, but behave quite differently upon kidney capsule transplantation to test their multipotency. Hypertrophic chondrocyte derived SSPCs were able to give rise to a fully mature bone ossicle, complete with marrow cavity and adipocytes, while marrow SSPCs from other sources could only generate bone.

## Results

### Hypertrophic chondrocytes give rise to osteoblasts and other marrow-associated cells

To validate the osteogenic fate of hypertrophic chondrocytes, we first utilized the *Col10a1CreERT2* mouse line crossed with a *Rosa26^{fs-tdTomato}* reporter (*Col10a1CreERT2; Rosa26^{fs-tdTomato}*). These mice were harvested at E16.5 following tamoxifen injections from E13.5-E15.5. Hypertrophic chondrocytes and their descendants found within the newly formed bone marrow cavity were tdTOMATO+ (*Figure 1a*). A number of these descendants co-expressed the osteoblast protein OCN (*Figure 1b–c*, white arrow); however, OCN-, tdTOMATO+ cells were also apparent within the marrow (*Figure 1b–c*, orange arrow). Additionally, we injected *Col10a1CreERT2; Rosa26^{fs-tdTomato}* mice with a single dose of tamoxifen (TM) at postnatal day 6 (P6) and harvested at 12 hr, 24 hr, and 2 weeks following TM (*Figure 1d–g*). At 12 hr following TM administration, specific expression of the tdTOMATO reporter was observed within hypertrophic chondrocytes and no cells within the marrow or surrounding tissues (*Figure 1d*). At 24 hr following TM administration, the tdTOMATO reporter could now be observed in a limited number of marrow-associated cells directly below the cartilage growth plate, as well as within hypertrophic chondrocytes (*Figure 1e*). At 2 weeks following TM administration, the tdTOMATO reporter had cleared from the hypertrophic chondrocytes and was now exclusively present in a number marrow-associated cells; some of those being at a significant distance from the growth plate (*Figure 1f*). In this distal region of the marrow, we also observed trabecular bone lining cells that co-labeled for tdTOMATO and OCN (*Figure 1g*; white arrows), suggesting that these cells had transitioned from hypertrophic chondrocytes to osteoblasts. Interestingly, other tdTOMATO-positive marrow-associated cells were not directly in contact with bone matrix and did not co-label with OCN (*Figure 1g*; orange arrow), suggesting the presence of another cell type or cell stage within the bone marrow that may be derived from hypertrophic chondrocytes. Additionally, by inducing recombination at both 2 months and 4 months of age and chasing for 2 and 4 weeks, respectively, we continued to observe contribution of tdTOMATO+ cells to the bone marrow (*Figure 1—figure supplement 1a, b*).

To get a better understanding of the proportionality of these hypertrophic chondrocyte-derived marrow and osteogenic cells, we performed similar analyses on constitutively active *Col10a1Cre; Rosa26^{fs-tdTomato}* mice. While tdTOMATO reporter expression was observed within hypertrophic chondrocytes and osteogenic cells associated with trabecular and endocortical bone of mice at P0

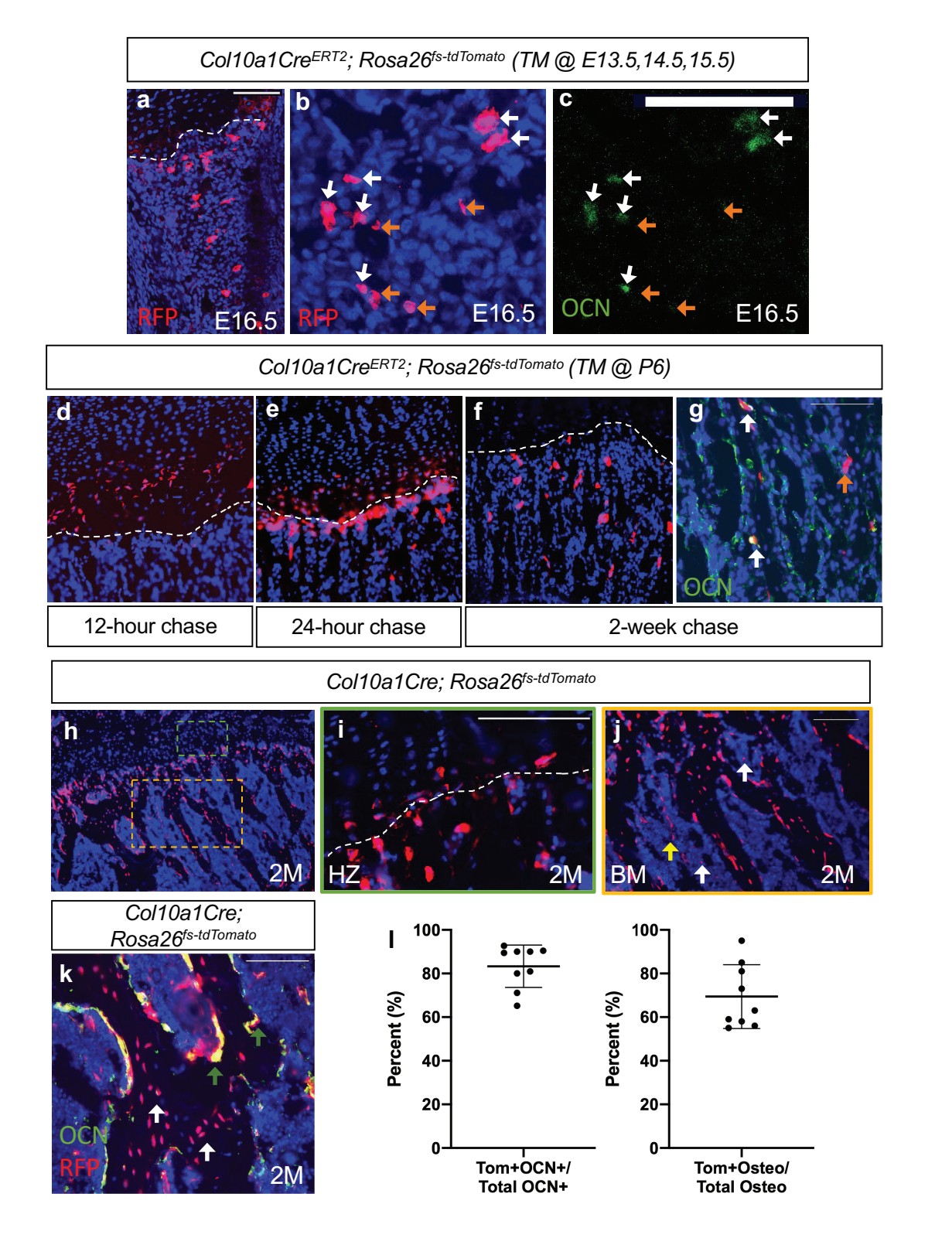

**Figure 1.** Hypertrophic chondrocytes are the primary source of osteoblasts/osteocytes in trabecular bone. Tibia sections of *Col10a1CreERT2; Rosa26fs-tdTomato* mice injected with tamoxifen at E13.5, 14.5, and 15.5 and subsequently sectioned at E16.5 and stained for (**a**) DAPI/RFP. (**b–c**) Higher magnification with OCN+/tdTOMATO+ descendants marked with white arrows and tdTOMTATO+ descendants not associated with bone (OCN-) are indicated by orange arrows. Tibia sections of *Col10a1CreERT2; Rosa26fs-tdTomato* mice injected with tamoxifen at P6 and subsequently sectioned at (**d**) 12 hour chase,

*Figure 1 continued on next page*

*Figure 1 continued*

(**e**) 24 hr chase, and (**f**) 2 week chase. (**g**) OCN immunostaining of *Col10a1CreERT2; Rosa26*<sup>fs-tdTomato</sup> bone sections following 2 week chase of TM. OCN⁺/tdTOMATO⁺ descendants marked with white arrows. tdTOMATO⁺ descendants not associated with bone (OCN⁻) are indicated by orange arrows. (**h**) Section of 2 M old *Col10a1Cre; Rosa26*<sup>fs-tdTomato</sup> tibia with higher magnifications of (**i**) hypertrophic zone (green box) and (**j**) bone marrow cavity (orange box). Non-bone lining TOMATO⁺ descendants marked with white arrows and potential vessel associated tdTOMATO⁺ descendants indicated by yellow arrow. (**k**) OCN and RFP immunostaining of 2 M *Col10a1Cre; Rosa26*<sup>fs-tdTomato</sup> tibia. OCN⁺/tdTOMATO⁺ osteoblasts represented by green arrows and tdTOMATO⁺ osteocytes with white arrows with quantifications shown in (l – *Figure 1—source data 1*). Scale bars = 100 um. N = 3 slides from three biologic replicates, SD ±9.7% for OCN stain, SD ±14.8% for osteocytes. Dotted line demarks the chondro-osseous junction.

The online version of this article includes the following source data and figure supplement(s) for figure 1:

**Source data 1.** Osteoblast and osteocyte quantification on Col10a1Cre;Rosa26fs-tdTomato 2M old mice.

**Figure supplement 1.** Contribution of hypertrophic chondrocytes to the bone marrow are observed into skeletal maturity and CRE expression is restricted to hypertrophic chondrocytes in *Col10a1Cre* mice.

(*Figure 1—figure supplement 1d-f*) and 2 months (2 M) of age (*Figure 1h–j*), CRE recombinase could only be detected within hypertrophic chondrocytes (*Figure 1—figure supplement 1c*). The vast majority of the trabecular and endocortical bone was lined with tdTOMATO reporter-positive cells, and 69.4% ± 14.8% of osteocytes were reporter positive (*Figure 1k and l* – white arrows). We also observed that 83.3% ± 9.7% of all OCN-positive osteoblasts were tdTOMATO reporter positive, confirming that hypertrophic chondrocytes can give rise to the majority of intramedullary osteoblasts during bone development (*Figure 1k and l* – green arrows). Of note, a number of tdTOMATO negative and OCN positive cells were also identified lining portions of the trabecular and endocortical bone, indicating that some of these osteoblasts originate from an alternate source such as the perichondrium/periosteum (*Figure 1k*). Finally, utilizing this model, we were able to confirm the existence of tdTOMATO reporter expressing cells within the marrow that were not associated with bone matrix (white arrow); many of which were localized to vessel-like structures (yellow arrow) (*Figure 1j*).

## Single-cell transcriptomics identifies several cell populations, including osteoblasts that are derived from hypertrophic chondrocytes during embryonic and postnatal development

Following the observation that tdTOMATO reporter-positive cells were both marrow- and bone-associated (*Figure 1*), we hypothesized that the hypertrophic chondrocytes may give rise to multiple cell fates. We therefore took an unbiased approach by performing single cell RNA-sequencing (scRNA-seq) on cells derived from *Col10a1Cre; Rosa26*<sup>fs-tdTomato</sup> mice at E16.5, a timepoint coincident with bone marrow cavity initiation and the potential transition of hypertrophic chondrocytes to alternative fates. To isolate the hypertrophic chondrocytes and their descendants we liberated the femur, tibia, and fibula from soft tissues and performed collagenase digestions of the skeletal rudiments. Following matrix digestion, red blood cells were lysed and the remaining cell populations were subjected to FACS. We selected tdTOMATO reporter positive cells that demonstrated high levels of tdTOMATO fluorescence for further analysis (*Figure 2—figure supplement 1a*). Using the 10 x Genomics platform, we performed scRNA-seq and further analyzed 2,273 tdTOMATO reporter positive sorted cells following quality control processing and elimination of a small number of cells lacking *tdTomato* transcripts. We then performed a Seurat-based unsupervised clustering of gene expression profiles from the tdTOMATO positive cells, which ultimately yielded eight distinct cell clusters (*Figure 2a*; see Materials and methods for details). Initial observations demonstrated that both *Cre* and *Col10a1* transcripts were detected in a portion of the cells, but we also observed cells expressing *tdTomato* without expression of either *Cre* or *Col10a1*, indicating the presence of both hypertrophic chondrocytes and their descendants within these clusters (*Figure 2—figure supplement 2a, i*).

Next we analyzed expression of chondrogenic genes. Differential gene expression analysis revealed the presence of *Ihh* and *Col10a1* expressing chondrocytes (most notably in clusters 1, 3, and 5), identifying cells in transition from pre-hypertrophy to hypertrophy (*Figure 2b and i*; *St-Jacques et al., 1999*). *Col10a1* expression was observed throughout many clusters; however was most highly enriched in clusters 1–5 and 7 (*Figure 2c and i*). Moderate expression of *Sox9* and *Aggrecan* were observed throughout all clusters, but primarily enriched in clusters 1–5 (*Figure 2—figure supplement 2b-c, i*). *Mmp13*, noted for its role in terminal hypertrophic chondrocytes, reached highest expression

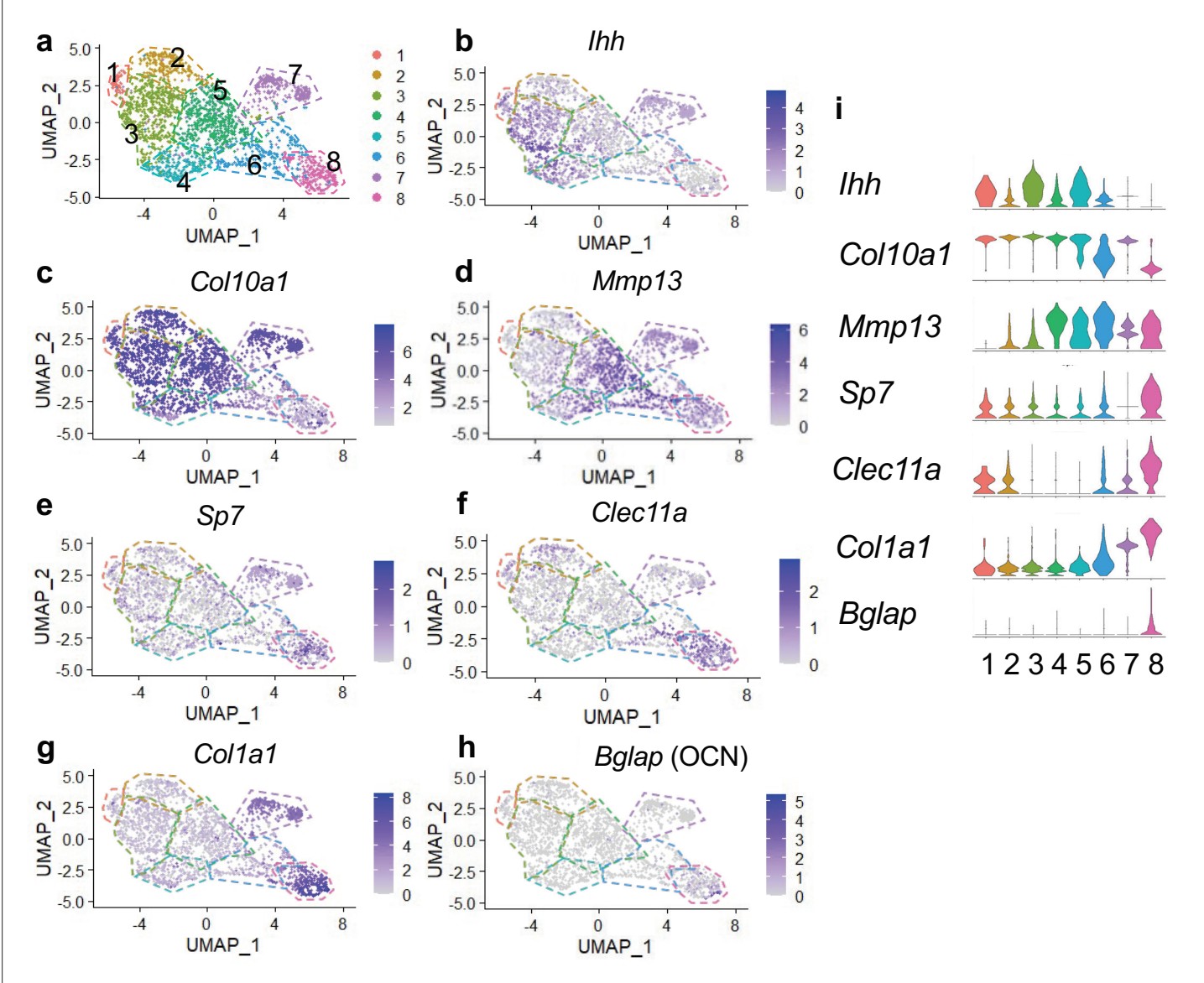

**Figure 2.** Single cell transcriptomics at E16.5 captures hypertrophic chondrocytes and osteoblasts. (**a**) UMAP shown in two-dimensional space produced using Seurat 4 package of R from single-cell RNA-sequencing of cleaned skeletal rudiments from *Col10Cre; Rosa26*$^{fs-tdTomato}$ mice at E16.5. (**b–d**) Feature plots of hypertrophic chondrocyte associated genes identified in clusters 1–7. (**e–h**) Feature plots of osteoprogenitor/osteoblast specific genes identified in clusters 7–8. (**i**) Violin plot representing the relative level of chondrocyte and osteoblast associated gene expression (**b–h**).

The online version of this article includes the following figure supplement(s) for figure 2:

**Figure supplement 1.** FACS plots for isolation of tdTOMATO⁺ cells at E16.5 and 2 M for scRNA-sequencing.

**Figure supplement 2.** Additional genes of interest expressed in hypertrophic chondrocytes and descendants at E16.5.

in clusters 4, 5, and 6 (***Figure 3d and i***). The osteogenic genes, *Sp7, Clec11a, Col1a1, Bglap, Runx2, Spp1, Alpl,* and *Ibsp,* were most highly expressed in clusters 7 and 8, which represent two potential osteoblast clusters (***Figure 2e–h and i*** and ***Figure 2—figure supplement 2d-i***), while some were also detected at much lower levels in pre-hypertrophic/hypertrophic chondrocyte clusters.

Differential gene expression analysis also identified the SSPC-associated genes, *Pdgfra, Ly6a* (encoding SCA1)*, Lepr, Cxcl12,* to be expressed within clusters 4, 5, and 6, with some expression in cluster 8 (***Figure 3a–e***). Each of these genes and/or their regulatory elements genetically identify SSPCs capable of becoming osteoblasts in vivo (***Crisan et al., 2008***; ***Morikawa et al., 2009***; ***Omatsu et al., 2010***; ***Worthley et al., 2015***; ***Zhou et al., 2014a***). We did not observe *Adipoq* expression in

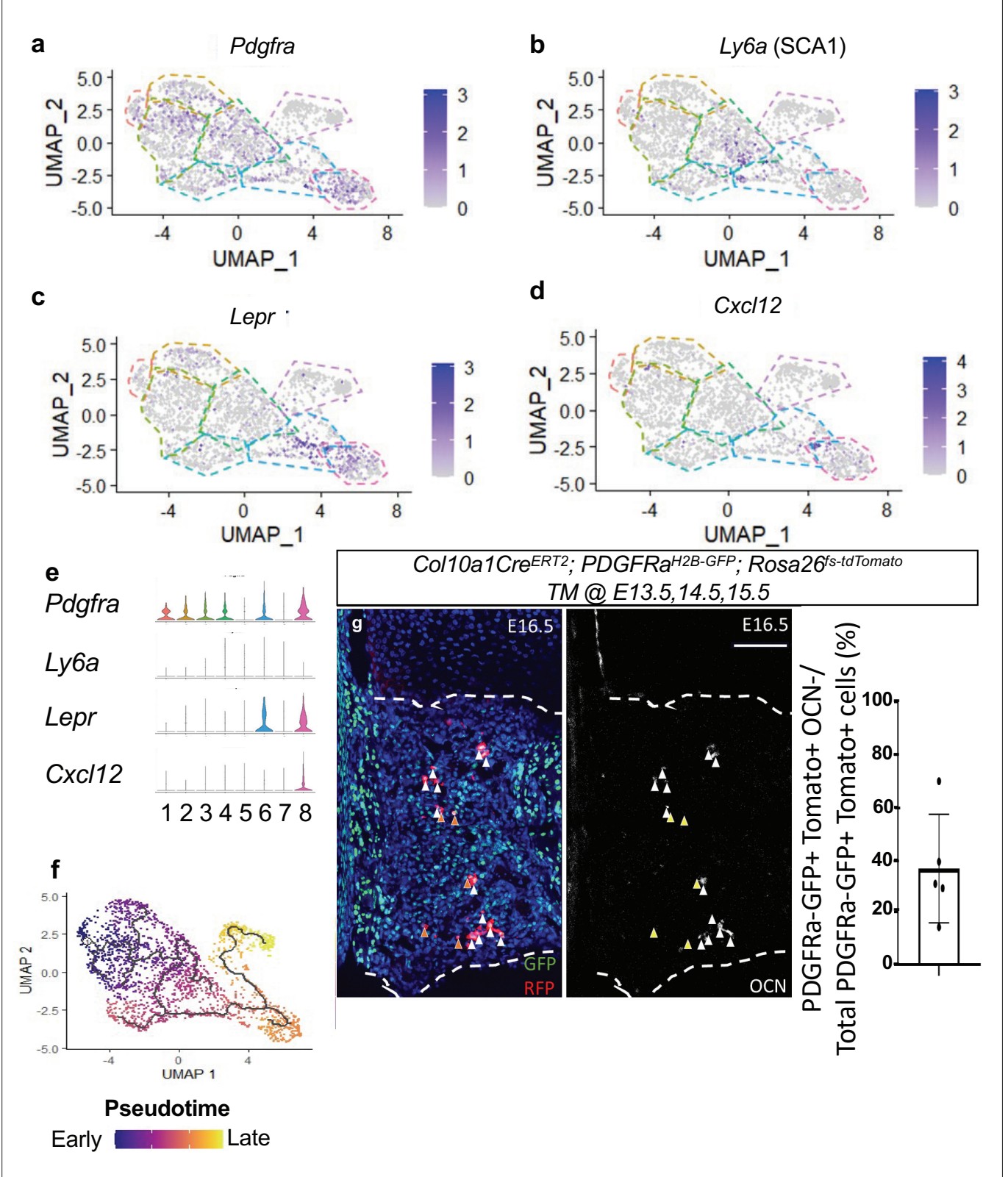

**Figure 3.** Single-cell transcriptomics of E16.5 hypertrophic chondrocytes and descendants reveals an intermediate SSPC upstream of osteoblasts. (a–d) Feature plot of SSPC-associated genes identified between hypertrophic chondrocyte and osteoblast clusters (e) Violin plot representing the relative level of SSPC associated gene expression (a–d). (f) Monocle three trajectory analysis throughout pseudotime. (g) E16.5 tibia sections of *Col10a1CreERT2; Rosa26^{fs-tdTomato};PDGFRa^{H2B-GFP}* mice injected with tamoxifen at E13.5, 14.5, and 15.5 and quantification (*Figure 3—source data 1*).

*Figure 3 continued on next page*

*Figure 3 continued*

Orange arrows represent tdTOMATO$^+$/PDGFRa$^{H2B-GFP+}$ cells that are OCN$^-$. White arrows represent tdTOMATO$^+$/ PDGFRa$^{H2B-GFP+}$ cells that co-express OCN. Scale bar = 100 um. N = 1/2 slides for three biological replicates, Average = 36.7%, SD ±20.7%. Dotted line demarks the chondro-osseous junction.

The online version of this article includes the following source data and figure supplement(s) for figure 3:

**Source data 1.** PDGFRa+ TOMATO+ quantification at e16.5 of Col10a1Cre; Rosa26-fs-tdTomato; PDGFRa-H2B-GFP.

**Figure supplement 1.** SSPC associated proteins are detected in hypertrophic chondrocyte-derived marrow-associated cells.

**Figure supplement 1—source data 1.** LEPR+ TOMATO+ quantification of Col10a1Cre;Rosa26fs-tdTomato at 2M.

**Figure supplement 1—source data 2.** Flow cytometric analysis of PDGFRa-H2B-GFP and tdTOMATO on Col10a1Cre;Rosa26fs-tdTomato at 2M.

**Figure supplement 1—source data 3.** Flow cytometric analysis of PDGFRb and tdTOMATO on Col10a1Cre;Rosa26fs-tdTomato at 2M.

any clusters, another gene previously shown to be co-expressed with *Lepr* in some SSPC populations (*Figure 2—figure supplement 2h-i*; *Zhong et al., 2020*). To further understand the cluster hierarchy of these hypertrophic chondrocytes and their descendants, we performed pseudotime analysis of the scRNA-seq data. This analysis indicated a singular trajectory linking hypertrophic chondrocytes (clusters 1–5) to osteoblasts (clusters 7 and 8) by way of an intermediate SSPC-like population (cluster 6) (*Figure 3f*). To validate this transition of hypertrophic chondrocytes to a marrow-associated SSPC fate, we generated *Col10a1CreERT2; Rosa26$^{fs-tdTomato}$; PDGFRa$^{H2B-GFP}$* mice and performed additional in vivo lineage-tracing analysis at E16.5. Fluorescent reporter analysis identified a number of tdTOMATO and PDGFRa$^{H2B-GFP}$ double positive cells within the marrow space that were either OCN$^+$ or OCN$^-$ (*Figure 3g–h*), indicating that tdTOMATO$^+$, GFP$^+$, and OCN$^-$ cells likely represent a more primitive hypertrophic chondrocyte-derived SSPC. To further validate that hypertrophic chondrocytes are capable of giving rise to marrow-associated SSPCs expressing *Lepr, Pdgfra,* and *Pdgfrb* in vivo throughout development, we performed IF co-labeling assays on bone sections from *Col10a1CreERT2; Rosa26-tdTomato $^{f/+}$* mice at 3 W of age (TM delivered at P6). Co-labeling of cells with tdTOMATO fluorescence and antibodies for LEPR, PDGFRα, and PDGFRβ demonstrate that hypertrophic chondrocytes indeed give rise to marrow associated SSPCs during skeletal development (*Figure 3—figure supplement 1a*).

After identifying *Col10a1CreERT2; Rosa26$^{fs-tdTomato}$* reporter positive SSPCs at early stages of development, we hypothesized that these cells may still arise from hypertrophic chondrocytes during later stages of skeletal development. Therefore, we performed scRNA-seq of tdTOMATO$^+$ sorted marrow cells from *Col10a1Cre; Rosa26$^{fs-tdTomato}$* mice at 2 months of age (2 M). To isolate and specifically enrich hypertrophic chondrocyte-derived marrow-associated cells, we performed centrifugation of the marrow followed by a gentle flush of femurs and tibias and subjected these marrow plugs to a brief collagenase digestion. This short digestion allowed for potential isolation of a limited number of osteoblasts, since our focus was centered on the analysis of novel non-osteoblast/osteocyte populations. Following the digest, red blood cells were lysed, and the remaining cell populations were subjected to FACS. We selected tdTOMATO reporter positive cells that demonstrated high levels of unambiguous tdTOMATO fluorescence for further analysis (*Figure 2—figure supplement 1b*). Using the 10 x Genomics platform, we performed scRNA-seq and further analyzed 1,066 tdTOMATO reporter positive sorted cells following quality control processing and elimination of any cells lacking *tdTomato* transcripts. Since neither *Cre* nor *Col10a1* expression could be detected in any of these cells, they clearly represented descendants of *Col10a1Cre*-expressing hypertrophic chondrocytes. Finally, we performed a Seurat-based unsupervised clustering of gene expression profiles from the tdTOMATO-positive cells, which ultimately yielded five distinct cell clusters (*Figure 4a*; see Materials and methods for details).

Differential gene expression analyses further demonstrated that cells associated with clusters 1 and 2 expressed several known osteogenic genes (*Figure 4b–h*). Many cells in cluster one expressed genes such as *Mepe, Dmp1,* and *Fam20c* (*Figure 4b–d and h*), known for their roles in osteocytes and/or the bone mineralization process (*Robling and Bonewald, 2020*; *Sapir-Koren and Livshits, 2014*). Some cells in cluster 2 also expressed *Dmp1* and *Fam20c*; however, their expression was detected at much lower levels. Most cells in cluster two expressed other osteoblast associated genes, such as *Bglap* and *Col1a1* (*Figure 4e–f and h*), critical components of the organic bone matrix. In addition to the few genes highlighted here, additional osteoblast-associated genes were also observed in cells within

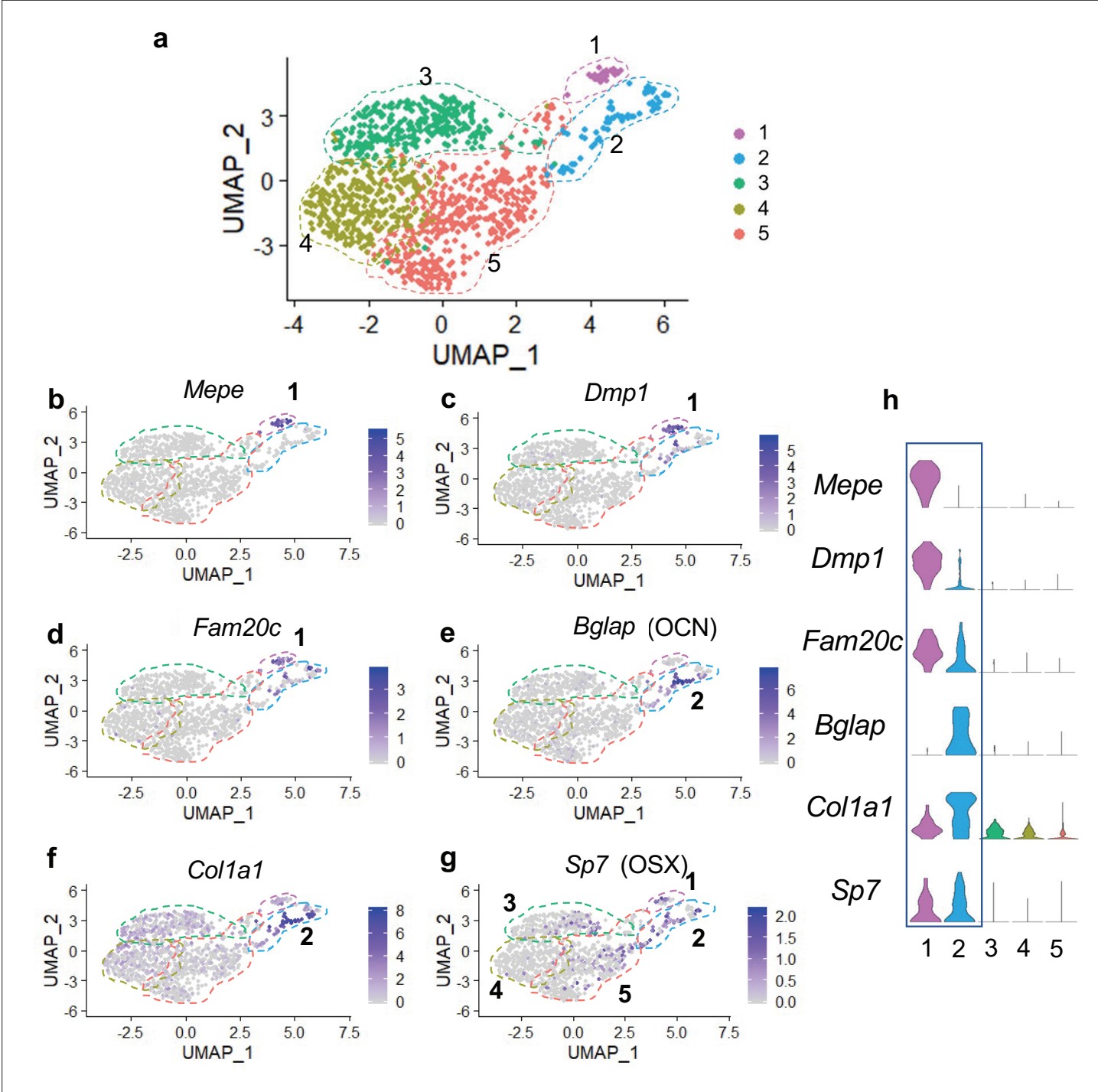

**Figure 4.** Single cell transcriptomics of hypertrophic chondrocyte descendants following FACS and 10 X Genomics sequencing at 2 M of age. (**a**) UMAP shown in two-dimensional space produced using Seurat 3 package of R from single-cell RNA-sequencing of bone marrow digest from *Col10Cre; Rosa26^fs-tdTomato^* mice at 2 M. (**b–f**) Feature plots of osteoblast specific genes identified in cluster 1 (**b–d**) and cluster 2 (**e–f**). (**g**) Feature plot of the osteoprogenitor associated gene, *Sp7*. (**h**) Violin plot representing the relative level of osteoblast-associated gene expression (**b–g**).

The online version of this article includes the following figure supplement(s) for figure 4:

**Figure supplement 1.** Additional osteoblast associated genes observed in clusters 1 and 2 at 2 months of age.

**Figure supplement 2.** Differentially expressed genes mostly unique to clusters 3, 4, and 5.

**Figure supplement 3.** Hypertrophic chondrocyte-derived cells can associate with blood vessels; however, do not express genes associated with pericytes or vascular smooth muscle cells.

*Figure 4 continued on next page*

*Figure 4 continued*

**Figure supplement 4.** *LeprCre; R26-tdTomato* and *AdipoqCre; R26-tdTomato* mice exhibit reporter expression in marrow associated cells, osteoblasts, and adipocytes with age.

clusters 1 and 2 (*Figure 4—figure supplement 1a-h*). Expression analyses further determined that the master transcriptional regulator of osteoblast differentiation and known osteoprogenitor cell marker, S*p7* (encoding the protein OSTERIX) (*Maes et al., 2010*; *Mizoguchi et al., 2014*), not only exhibited expression within osteoblast clusters 1 and 2, but also within a limited number of cells associated with clusters 3, 4, and 5 (*Figure 4g–h*). These data suggest that the majority of marrow-associated cells derived from the hypertrophic chondrocytes localized to clusters 3, 4, and 5 are not osteoblasts or osteoprogenitors, but rather may be more primitive cells within the osteoblast lineage or distinct cell types altogether. We next utilized differential gene expression analyses to identify genes that were uniquely and broadly expressed across the cells of clusters 3, 4, 5, with limited expression in the more differentiated osteoblasts of clusters 1 and 2. This analysis identified a number of genes utilized for identifying SSPC populations. Specifically, *Lepr, Grem1, Pdgfra* (but not *Ly6a* – SCA1), *Cxcl12*, and *Pdgfrb* were highly expressed within cells across all three clusters (*Figure 5a–f and k*). Each gene and/ or their regulatory elements genetically identify or can be utilized to label SSPCs with the potential of becoming osteoblasts in vivo (*Crisan et al., 2008*; *Morikawa et al., 2009*; *Omatsu et al., 2010*; *Worthley et al., 2015*; *Zhou et al., 2014a*). To confirm that hypertrophic chondrocytes are capable of giving rise to marrow associated SSPCs expressing *Lepr*, *Pdgfra*, and *Pdgfrb* in vivo at this later developmental time point, we performed IF co-labeling assays on bone sections from *Col10a1Cre; Rosa26$^{fs-tdTomato}$* mice at 2months of age. Co-labeling of cells with tdTOMATO fluorescence and antibodies for LEPR, PDGFRα, and PDGFRβ further demonstrate and validate that hypertrophic chondrocytes indeed give rise to marrow-associated SSPCs during postnatal skeletal development (*Figure 3—figure supplement 1b*). Quantification of *Col10a1Cre; Rosa26$^{fs-tdTomato}$* for LEPR demonstrates that 74.21%±13.92% of all tdTOMATO+ cells were LEPR+ (*Figure 3—figure supplement 1c*). We additionally performed flow cytometry of marrow cells from *Col10a1Cre; Rosa26$^{fs-tdTomato}$; PDGFRa$^{H2B-GFP}$* mice at 1 month of age and observed 73.6% ± 2.83% of tdTOMATO$^+$ cells were PDGFRa-GFP+ (*Figure 3—figure supplement 1c*). At 2 M of age, flow cytometry of marrow cells from *Col10a1Cre; Rosa26$^{fs-tdTomato}$* mice demonstrated that 64.33% ± 3.48% of tdTOMATO$^+$ cells were PDGFRB+ (*Figure 3—figure supplement 1c*) and that 37.4% ± 6.2% of the total LEPR$^+$ population were also tdTOMATO$^+$ as per quantitation from stained bone sections (*Figure 5—figure supplement 1a*). This was further corroborated by utilizing flow cytometry of marrow cells from *Col10a1Cre; Rosa26$^{fs-tdTomato}$; PDGFRa$^{H2B-GFP}$* mice, which demonstrates that a similar 26.93%±3.79% of all PDGFRa$^{H2B-GFP+}$ cells were also tdTOMATO reporter positive (*Figure 5—figure supplement 1b*). These data indicate that hypertrophic chondrocyte-derived SSPCs constitute nearly 25–45% of all marrow SSPCs at these stages, and that a significant portion of the marrow-associated SSPCs are likely derived from other origins such as the perichondrium and/or periosteum in mice 2 months of age and younger.

FACS has been highly utilized to identify, isolate, and enrich populations of SSPCs from bone marrow and periosteal tissues in mice and/or humans based on the expression of specific cell surface proteins (*Chan et al., 2015*; *Chan et al., 2018*; *Zhou et al., 2014a*). FACS assays utilizing antibodies against Integrin Subunit Alpha V (ITGAV) have been used for the identification, isolation, and enrichment of bone marrow stromal cell populations in mice, which are noted to be largely quiescent and capable of generating most of the colonies within a colony-forming unit – fibroblastic (CFU-F) assay; a technique utilized to identify heterogeneous, non-hematopoietic, marrow stromal cells with progenitor cell characteristics (*Boulais et al., 2018*; *Robey et al., 2014*). While we identified *Itgav* expression sporadically throughout cells of all clusters, it is enriched in the SSPC clusters 3, 4, and 5 (*Figure 5g and k*). Recently, CD200, ITGAV, and CD105 antibodies have been used in conjunction with antibodies detecting the hematopoietic and endothelial lineages (CD45, CD31, and Ter119) for flow cytometric evaluation and identification of unique subsets of SSPCs (*Gulati et al., 2018*). These studies indicate that lineage negative (CD45$^-$,CD31$^-$,Ter119$^-$) and CD200$^+$,ITGAV$^+$,CD105$^-$ cells represent the most primitive self-renewing and multipotent skeletal stem cell (SSC) population, while lineage negative and ITGAV$^+$,CD105$^+$ cells represent a more committed SSPC population (*Gulati et al., 2018*). Our gene expression analyses observed broad expression of *Cd200* across all clusters including the osteoblasts; however, *Eng* (encoding CD105) was more restricted to SSPC clusters 3, 4, 5 and enriched

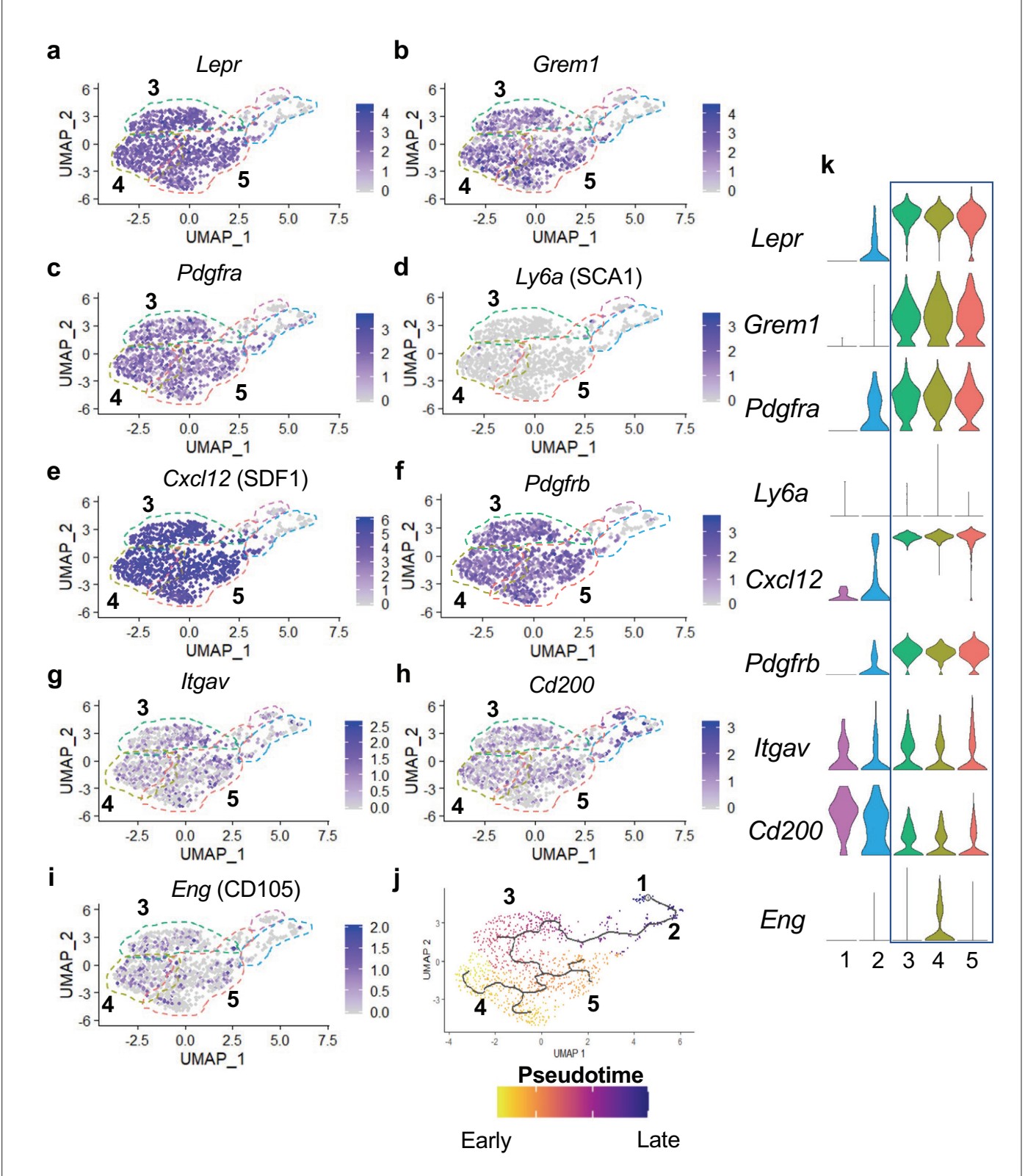

**Figure 5.** Many hypertrophic chondrocyte derived cells express genes associated with SSPCs. (**a–f**) Feature plots of genes previously identified as SSPC markers in genetic mouse models. (**g–i**) Feature plots of genes previously identified as SSPC markers for use in flow cytometry and FACS. (**j**) Monocle three trajectory analysis with clusters noted throughout pseudotime. (**k**) Violin plots of SSPC-associated genes in (**a–i**).

*Figure 5 continued on next page*

*Figure 5 continued*

The online version of this article includes the following source data and figure supplement(s) for figure 5:

**Figure supplement 1.** Contribution of hypertrophic chondrocyte descendants to total SSPC populations.

**Figure supplement 1—source data 1.** LEPR+ TOMATO+ quantification on Col10a1Cre;Rosa26fs-tdTomato at 2M.

**Figure supplement 1—source data 2.** Flow cytometric analysis of PDGFRa-H2B-GFP and TOMATO on Col10a1Cre;Rosa26fs-tdTomato at 2M.

specifically within cluster 4 (*Figure 5h–i and k*). When assessing the precise cellular overlap of these factors, we observed 128 of 953 SSPCs to be *Itgav+, Cd200+, Cd105-*. While this most primitive SSC population exists among our SSPCs analyzed via scRNA-seq, they were relatively evenly distributed among cells within clusters 3, 4, and 5 (*Figure 5g–l and k*). These data suggest that hypertrophic chondrocytes likely undergo a process of dedifferentiation to generate even the most primitive SSCs; however, these cells themselves may be heterogeneous due to the differential expression of genes that allow for their segregation into multiple distinct clusters within our Seurat analysis.

In an attempt to further delineate each of the SSPC clusters in our Seurat analysis, we analyzed the data to identify genes restricted in their expression to a single SSPC cluster. A limited number of genes were ultimately found to be uniquely expressed within solitary SSPC clusters; however, much more commonly we observed genes to be enriched within specific clusters with limited expression within the other one or two SSPC clusters (*Figure 4—figure supplement 2*).

SSPCs are known to reside within specific niches often associated with blood vessels (*Zhou et al., 2014a*). Previously, we noted that some of the hypertrophic chondrocyte derived tdTOMATO reporter cells appeared to associate with vascular structures. To assess this directly, we performed immunofluorescent staining for ENDOMUCIN (EMCN) on bone sections of *Col10a1Cre; Rosa26*<sup>fs-tdTomato</sup> mice at 2 months of age and identified a number of tdTOMATO+ reporter cells directly adjacent to EMCN expressing endothelial cells (*Figure 4—figure supplement 3a*). To ensure that these tdTOMATO+ cells were not pericytes or vascular smooth muscle cells (vSMCs), we assessed our differential gene expression data for genes that serve as common markers for these specific cell types, including *Acta2, Nes, Kcnj8, Rgs5, Cspg4,* and *Mcam* (*Chasseigneaux et al., 2018*; *Yamazaki and Mukouyama, 2018*). We observed limited or no gene expression for these markers in SSPCs of clusters 3, 4, or 5 (*Figure 4—figure supplement 3b-g*).

To finally parse the relatedness or relationship between SSPC and osteoblast cell clusters, we performed Monocle three pseudotime cell trajectory analysis of the Seurat data. These data indicate a clear hierarchy of cellular differentiation and relatedness between all cell clusters in an unbranched or continuous trajectory. Clusters 4 and 5 shared a common trajectory, with cluster 4 being slightly earlier in pseudotime. These clusters then give rise to cluster 3, which further differentiates into the osteoblast clusters 2 and then 1 (*Figure 5j*). Taken together these data indicate that cluster 3, 4, and 5 are hypertrophic chondrocyte-derived SSPCs that are capable of becoming osteoblasts and have unique niche localizations with at least some of these SSPCs being vessel associated.

## Hypertrophic chondrocyte-derived SSPCs are dually primed for both osteoblast and adipocyte differentiation

Since most bone-marrow-associated SSPCs have the potential to become either osteoblasts or adipocytes during development, aging, regeneration, and/or disease, we further examined our differential gene expression data for indicators of this competency. While we observed limited expression of the osteoprogenitor transcription factor *Sp7* in the SSPC clusters 3, 4, and 5, we noticed much broader and robust expression of *Runx2,* the earliest master transcriptional regulator of osteoblast specification, throughout all clusters and especially concentrated in the SSPC clusters 3, 4, and 5 (*Figure 6a*). RUNX2 has been observed in LEPR<sup>+</sup> cells and appears to prime cells for osteoblastic specification and differentiation (*Yang et al., 2017*). We have already demonstrated in *Figure 1* the efficiency by which hypertrophic chondrocytes can contribute to the osteoblast lineage, and that these osteoblasts are part of a differentiation trajectory that can originate from hypertrophic chondrocyte-derived SSPCs.

Recent data has shown that some LEPR<sup>+</sup> SSPCs also express PPARγ (encoded by *Pparg*) and ADIPONECTIN (encoded by *Adipoq*) (*Zhong et al., 2020*). Similar to this report we observed *Pparg* and *Adipoq* expression within *Lepr*-expressing SSPCs of clusters 3, 4, and 5 (*Figure 6b–c*). While PPAR-gamma is necessary and sufficient to induce adipogenesis of SSPCs, ADIPONECTIN enhances

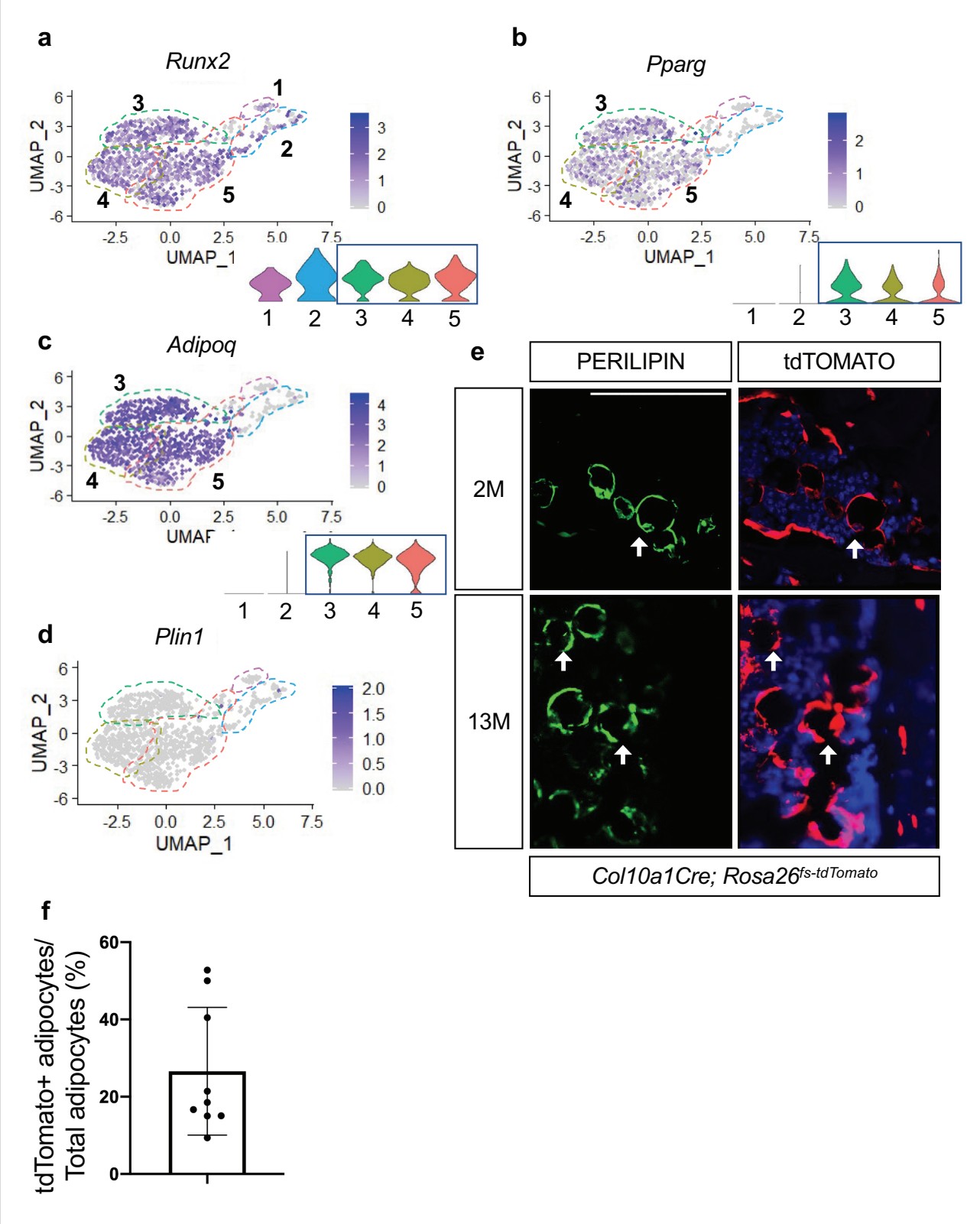

**Figure 6.** Hypertrophic chondrocyte derived SSPCs exhibit osteogenic and adipogenic differentiation capacities. (**a**) Feature plot and violin plot of the osteoblast specification gene, Runx2. (**b–c**) Feature plots and violin plots of the adipogenic specification gene, Pparg, and adipogenic associated gene, Adipoq. (**d**) Feature plot indicating a lack of expression of the mature lipid laden adipocyte gene, Perilipin. (**e**) Immunostaining for PERILIPIN in *Col10a1Cre;R26-tdTomato* mice at 2 months and 13 months of age. PERILIPIN+, tdTOMATO+ adipocytes noted with white arrows. Scale bar = 100 µm

*Figure 6 continued on next page*

*Figure 6 continued*

(**f**) Quantification of PERILIPIN⁺, tdTOMATO⁺ adipocytes at 13 months of age in (**e**) Average = 27.59% (*Figure 6—source data 1*). N = 3 slides from three biologic replicates, SD ±16.2%.

The online version of this article includes the following source data for figure 6:

**Source data 1.** PERILIPIN+ TOMATO+ quantifications on Col10a1Cre;Rosa26fs-tdTomato at 2M.

osteogenesis but is primarily secreted by adipocytes (*Tencerova and Kassem, 2016*). Given the expression of these adipogenic regulators and marker genes, we asked if hypertrophic chondrocyte-derived cells in clusters 3, 4, and 5 represent mature adipocytes or whether these cells could ultimately give rise to mature adipocytes in vivo. Our differential gene expression analysis did not indicate the presence of *Perilipin* (*Plin1*), a mature lipid laden adipocyte marker, in SSPC clusters 3, 4, or 5, suggesting that none of these cells isolated utilizing our procedures were representative of mature adipocytes (*Figure 6d*). Providing evidence for an ultimate adipogenic fate for some of these SSPCs, *Col10a1Cre; Rosa26ᶠˢ⁻ᵗᵈᵀᵒᵐᵃᵗᵒ* mice at 2months and 13months of age displayed co-localization of the tdTOMATO reporter and PERILIPIN antibody staining (*Figure 6e*). By 13 M of age, approximately 25% of adipocytes labeled with PERILIPIN were also tdTOMATO reporter positive (*Figure 6f*). None of these double-labeled cells are either known to express *Col10a1*, nor could we detect CRE expression within any of these cells. Therefore, our results suggest that hypertrophic chondrocyte-derived SSPCs are also a significant contributor to the adipogenic fate; however, these mature cells likely eluded our scRNA-seq analyses due to the technical nature for isolating and sorting lipid laden adipocytes and their limited availability in the bone marrow at 2 months of age.

To assess SSPC contribution to the osteoblast and adipocyte lineages, we also performed reporter studies utilizing *LeprCre; Rosa26ᶠˢ⁻ᵗᵈᵀᵒᵐᵃᵗᵒ* and *AdipoqCre; Rosa26ᶠˢ⁻ᵗᵈᵀᵒᵐᵃᵗᵒ* mice. Utilizing bone sections of P0 *LeprCre; Rosa26ᶠˢ⁻ᵗᵈᵀᵒᵐᵃᵗᵒ* reporter mice, we observed reporter positive signals in both-marrow-associated cells and some bone lining cells; however, OCN⁺, tdTOMATO⁺ osteoblasts were not apparent (*Figure 4—figure supplement 4a*). By 2 M of age we observed bone lining cells, OCN-expressing osteoblasts, and osteocytes that were tdTOMATO reporter positive (*Figure 4—figure supplement 4a* – white arrows), similar to data previously reported (*Zhou et al., 2014a*). As expected, a limited number of PERILIPIN⁺, tdTOMATO⁺ adipocytes were also observed within the bone marrow of 2 M old *LeprCre; Rosa26ᶠˢ⁻ᵗᵈᵀᵒᵐᵃᵗᵒ* mice (*Figure 4—figure supplement 4a*). Similarly, P0 *AdipoqCre; Rosa26ᶠˢ⁻ᵗᵈᵀᵒᵐᵃᵗᵒ* bone sections exhibited marrow-associated and bone lining reporter-positive cells that not yet contributed to OCN+ osteoblasts; however, by 2 M of age tdTOMATO⁺ and OCN⁺ osteoblasts were observed, as well as tdTOMATO⁺ osteocytes and adipocytes (PLIN1+) were apparent in *Adipoq Cre; Rosa26ᶠˢ⁻ᵗᵈᵀᵒᵐᵃᵗᵒ* bone sections (*Figure 4—figure supplement 4b* – white arrows). In contrast with these studies, *Col10a1Cre; Rosa26ᶠˢ⁻ᵗᵈᵀᵒᵐᵃᵗᵒ* descendants demonstrate a relatively quick commitment to the osteoblast lineage that is observed at P0 and earlier, while *Col10a1CreERT2; Rosa26ᶠˢ⁻ᵗᵈᵀᵒᵐᵃᵗᵒ* mice document that bone lining cells and mature OCN-expressing osteoblasts can be generated from hypertrophic chondrocytes within 1 day to 2 weeks (*Figure 1a–g*). This highlights the potential for functional differences between sub-populations of LEPR and ADIPOQ expressing SSPCs based on their particular origin and/or niche.

## *Hypertrophic* chondrocyte-derived SSPCs may be uniquely primed for developmental bone growth in vivo

*Col10a1Cre*-derived marrow-associated SSPCs constitute only a portion of LEPR⁺ SSPCs and *LeprCre*-derived marrow-associated SSPCs contribute to nearly 100% of all CFU-F colonies, therefore we reasoned that *Col10a1Cre*-derived marrow-associated SSPCs would contribute to only a fraction of total CFU-Fs and that this may provide a method for separating hypertrophic chondrocyte-derived progenitors from those derived from alternative sources such as the perichondrium/periosteum (*Zhou et al., 2014a*). Following plating at clonal density and 10 days of growth, 50–60% of CFU-F colonies derived from *Col10a1Cre; Rosa26ᶠˢ⁻ᵗᵈᵀᵒᵐᵃᵗᵒ* mice exhibited relatively homogeneous tdTOMATO reporter expression while the other colonies were largely devoid of tdTOMATO positivity at both 2 months and 8 months of age (*Figure 7a–c*). To achieve a better understanding of the similarities and differences between progenitors derived from these different sources at the transcriptomic level, we isolated three tdTOMATO⁺ and three tdTOMATO⁻ CFU-F colonies and performed individual RNA-isolations and

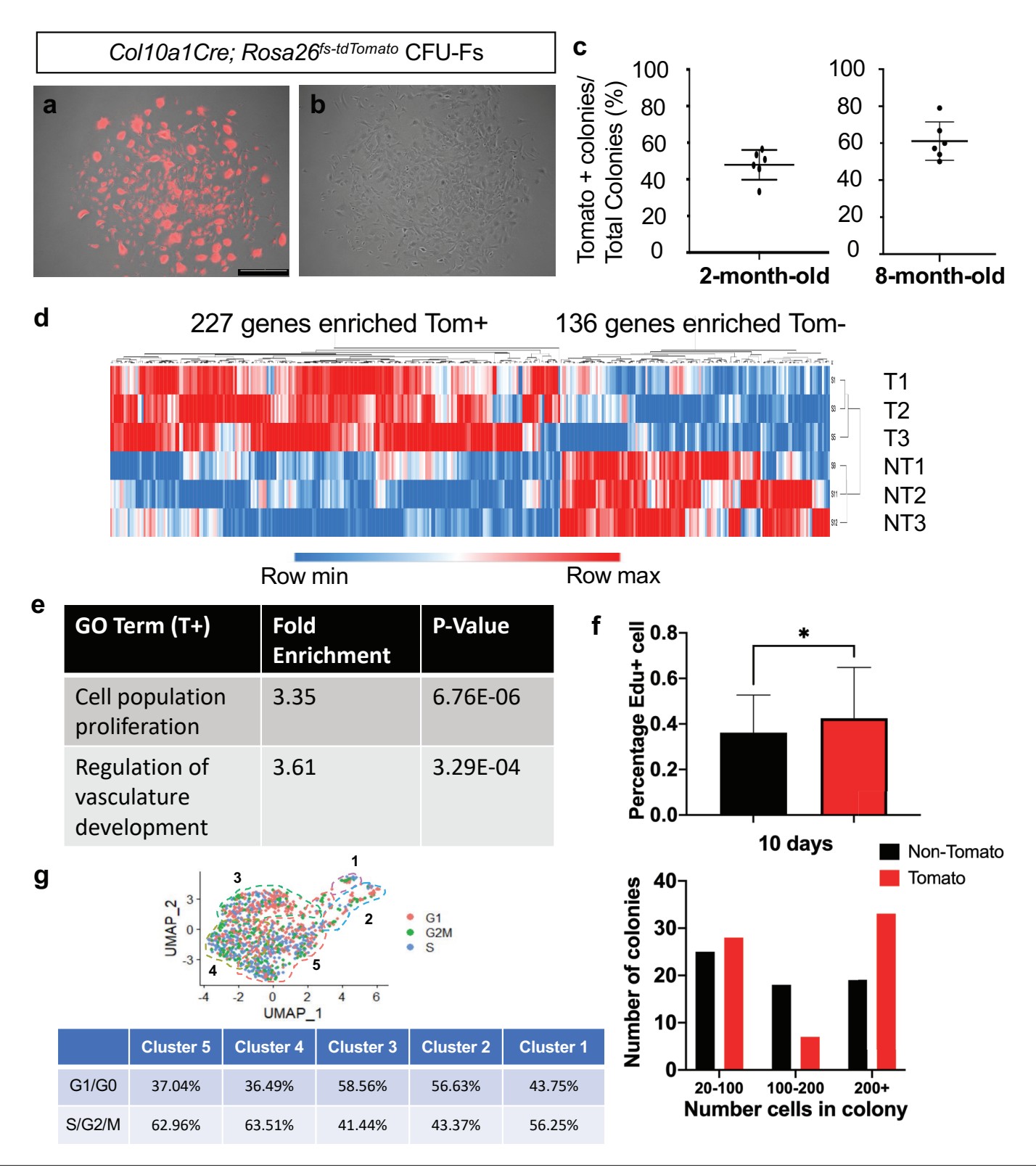

**Figure 7.** CFU-Fs derived from hypertrophic chondrocytes are similar to CFU-Fs derived from other cell sources, but contain SSPCs with enhanced proliferative capacities. (**a–c**) CFU-Fs derived from hypertrophic chondrocytes (tdTOMATO⁺) (**a**) and those derived from other cell sources (tdTOMATO⁻) (**b**) are established and develop at similar frequencies at 2- and 8 months of age (c – *Figure 7—source data 1*). Scale bar = 500 μm (**d**) Heat-map from bulk RNA-seq data from three tdTOMATO⁺ CFU-Fs and 3 TOMATO⁻ CFU-Fs (*Figure 7—source data 2*). (**e**) Gene ontology terms associated with

*Figure 7 continued on next page*

*Figure 7 continued*

tdTOMATO⁺ CFU-Fs (***Figure 7—source data 3*** and ***Figure 7—source data 4***). (**f**) EdU incorporation within tdTOMATO⁺ and tdTOMATO⁻ CFU-Fs (top - ***Figure 7—source data 5***) and differences in cell numbers between tdTOMATO⁺ and tdTOMATO⁻ CFU-Fs (bottom - ***Figure 7—source data 6***). N = 2 technical replicates of three biologic replicates, SD tdTOMATO⁻ ± 16.5%, SD tdTOMATO⁺ ± 22.3%, p-value = 0.044. (**g**) Cell cycle analysis of the SSPCs associated with clusters 1–5 using Seurat 3.

The online version of this article includes the following source data for figure 7:

**Source data 1.** Tomato+ colony unit formation quantifications at 2 and 8M of Col10a1Cre;Rosa26fs-tdTomato.

**Source data 2.** Bulk RNA-sequencing total gene list of 3 TOMATO+ and 3 TOMATO- colonies from Col10a1Cre;Rosa26fs-tdTomato.

**Source data 3.** Gene ontology analysis of genes enriched in TOMATO+ colonies.

**Source data 4.** Gene ontology analysis of genes enriched in TOMATO- colonies.

**Source data 5.** EDU+ quantifications of TOMATO+ and TOMATO- colonies.

**Source data 6.** Colony size of TOMATO+ and TOMATO- colonies.

submitted for bulk RNA-sequencing. Of the differentially expressed genes reaching significance (p < 0.05, LogFC >1), the three tdTOMATO⁺ CFU-Fs clustered separately from three tdTOMATO⁻ CFU-Fs on the basis of only 363 differentially expressed genes (227 enriched in tdTOMATO⁺ CFU-Fs and 136 enriched in tdTOMATO⁻ CFU-Fs) out of approximately 12,000 genes with an average expression level above biological thresholds (***Figure 7d*** and Methods for threshold calculations). We observed *Lepr, Grem1, Pdgfra, Adipoq, Pparg, Itgav, Cd200,* and *Cxcl12,* all highly expressed above biological thresholds in all CFU-Fs, indicating a general maintenance of SSPC phenotype while in culture and a striking similarity between progenitors from disparate sources.

To more readily identify differences between tdTOMATO⁺ and tdTOMATO⁻ CFU-Fs, we performed gene ontology analysis for biological processes and found two interesting terms arise due to enriched expression of genes within the tdTOMATO⁺ or hypertrophic chondrocyte derived CFU-Fs; cell population proliferation and regulation of vascular development (***Figure 7e***; ***Mi et al., 2019***; ***Gene Ontology Consortium, 2021***; ***Ashburner et al., 2000***). Regulation of vascular development was interesting, although not a surprising finding, since we know that terminal hypertrophic chondrocytes secrete factors to initiate vessel invasion from the periosteum and we had identified a number of marrow-associated hypertrophic chondrocyte derived SSPCs localized near blood vessels. The fact that the tdTOMATO⁺ or hypertrophic chondrocyte derived CFU-Fs were enriched in genes associated with cellular proliferation was unanticipated. To this end, we investigated EdU incorporation into all CFU-Fs from *Col10a1Cre; Rosa26ᶠˢ⁻ᵗᵈᵀᵒᵐᵃᵗᵒ* mice at day 5 in culture that consisted of at least 5 cells. We observed a greater percentage of cells in tdTOMATO⁺ CFU-Fs that had incorporated EdU as compared to tdTOMATO⁻ CFU-Fs (***Figure 7f*** - top). By day 10 in culture, tdTOMATO⁺ CFU-Fs on average were larger and displayed more colonies with 200+ cells as compared to tdTOMATO⁻ CFU-Fs (***Figure 7f*** - bottom). To determine whether this proliferative advantage could be observed in SSPCs immediately harvested from *Col10a1Cre; Rosa26ᶠˢ⁻ᵗᵈᵀᵒᵐᵃᵗᵒ* bone marrow, we performed cell cycle analysis on our scRNA-seq dataset and found that as a percentage of the population there was an equal or greater percentage of tdTOMATO⁺ SSPCs (cluster 3, 4, and 5) in a proliferative state (S, G2, or M phase) as compared to osteoblasts (clusters 1 and 2) (***Figure 7g***). This unexpectedly high proliferative capacity for hypertrophic chondrocyte-derived SSPCs may be reflective of the fact that hypertrophic chondrocytes eventually re-enter the cell cycle as they transition fates within the marrow space (***Park et al., 2015***; ***Yang et al., 2014a***).

To determine whether tdTOMATO⁺ and tdTOMATO⁻ CFUs contained inherent differences in their functional capacities to undergo osteogenic and adipogenic differentiation in vitro, we performed CFU-osteoblastic (CFU-Ob) and CFU-adipogenic (CFU-Ad) assays. Colonies composed of cells from either hypertrophic or non-hypertrophic origins were able to differentiate at similar capacities in vitro when measured by Alkaline Phosphatase, Von Kossa, and Oil Red O staining (***Figure 8—figure supplement 1***). However, when we performed kidney capsule transplantations to generate bone ossicles in vivo, there was a much more apparent functional difference between tdTOMATO⁺ and tdTOMATO⁻ cells. The tdTOMATO⁺ cells were able to generate a full ossicle with bone marrow, vessels, and adipocytes (***Figure 8a*** -top and ***Figure 8b***). Conversely, the tdTOMATO⁻ cells could only generate bone with no vascular invasion or adipocyte generation (***Figure 8*** -bottom and ***Figure 8b***). These apparent

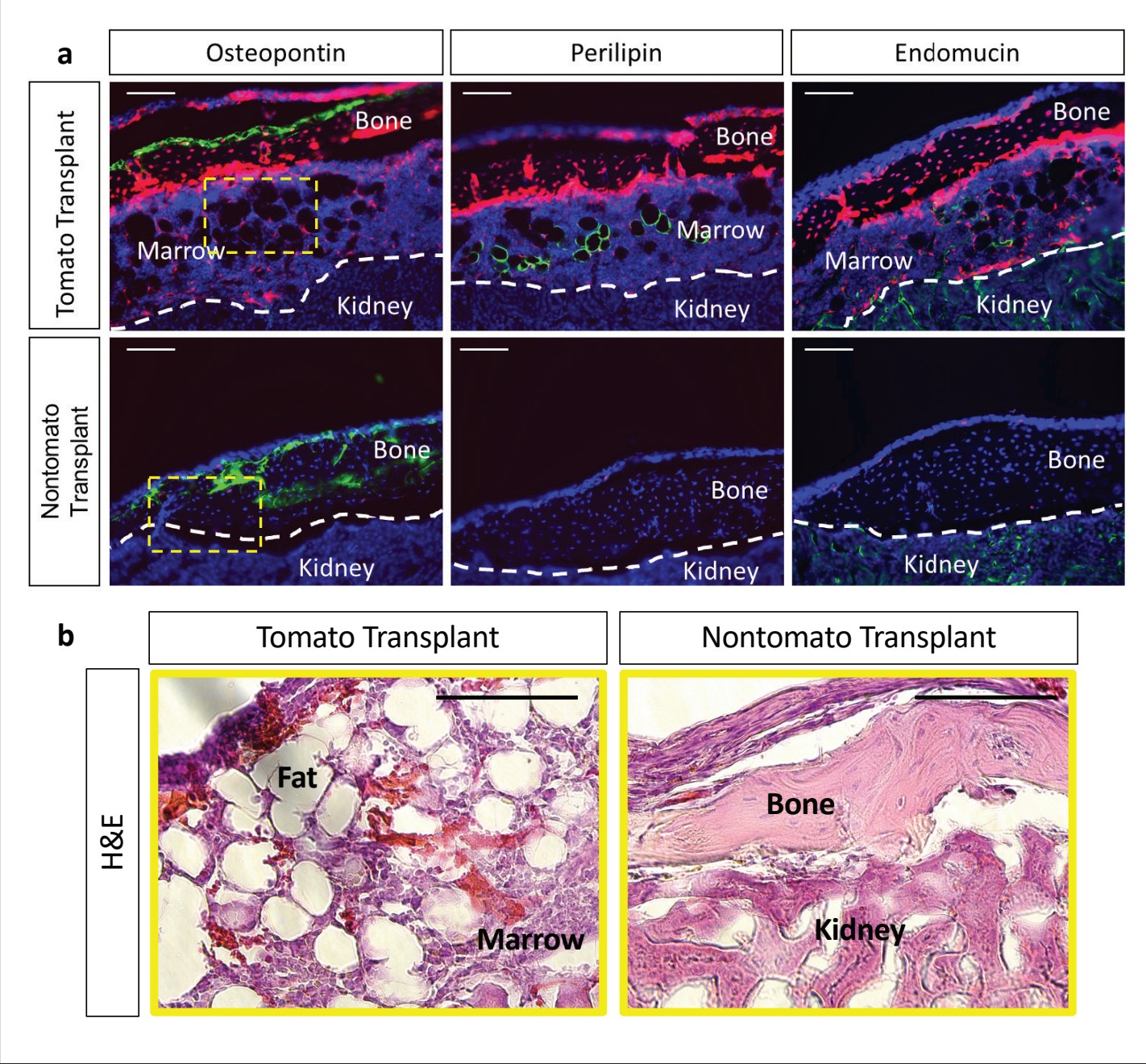

**Figure 8.** Kidney capsule transplantations of tdTOMATO+ cells exhibit complete ossicle formation with bone, adipocytes, and bone marrow compared to tdTOMATO- resulting in only bone formation. (**a**) Immunofluorescent stain for bone (osteopontin), adipocytes (perilipin), and vessels (endomucin) for tomato+ transplant (top) and tomato- transplant (bottom). (**b**) Hematoxylin and Eosin stain for marrow establishment in tdTomato+ transplant and tdTOMATO- transplant.

The online version of this article includes the following source data and figure supplement(s) for figure 8:

**Figure supplement 1.** In vitro differentiation of colonies reveals relatively similar osteogenic and adipogenic differentiation capacities.

**Figure supplement 1—source data 1.** Differentiation assay quantifications of TOMATO+ and TOMATO- colonies.

differences when transplanted suggest that hypertrophic chondrocyte-derived SSPCs are much more supportive of vascular invasion and marrow formation than SSPCs derived from other cell sources.

## Discussion

Recent and historic data have suggested that hypertrophic chondrocytes are capable of becoming osteoblasts that directly contribute to trabecular bone formation (*Giovannone et al., 2019*; *Park et al., 2015*; *Tsang et al., 2015*; *Yang et al., 2014b*; *Yang et al., 2014a*; *Zhou et al., 2014b*); however, the process by which hypertrophic chondrocytes generate osteoblasts remains unknown. Current genetic lineage tracing studies have suggested a transdifferentiation mechanism by which hypertrophic chondrocytes differentiate directly into an osteoprogenitor or mature osteoblast without transiting through an intermediate or more primitive state (*Park et al., 2015*). Here, we have provided genetic and transcriptomic evidence, consistent with our hypothesis, that hypertrophic chondrocytes undergo a process of dedifferentiation to form SSPC populations ultimately giving rise to osteoblastic and adipogenic cell fates within the marrow compartment of embryonic, postnatal, and adult bones. A number of genetic and flow cytometric approaches have been used to identify and isolate multipotent SSPC populations within both the bone marrow and periosteal compartments. Many of these efforts have used alternative genetic models (*Gremlin1Cre*, *LeprCre*, *Col2a1Cre*, *Cxcl12Cre*) with lineage

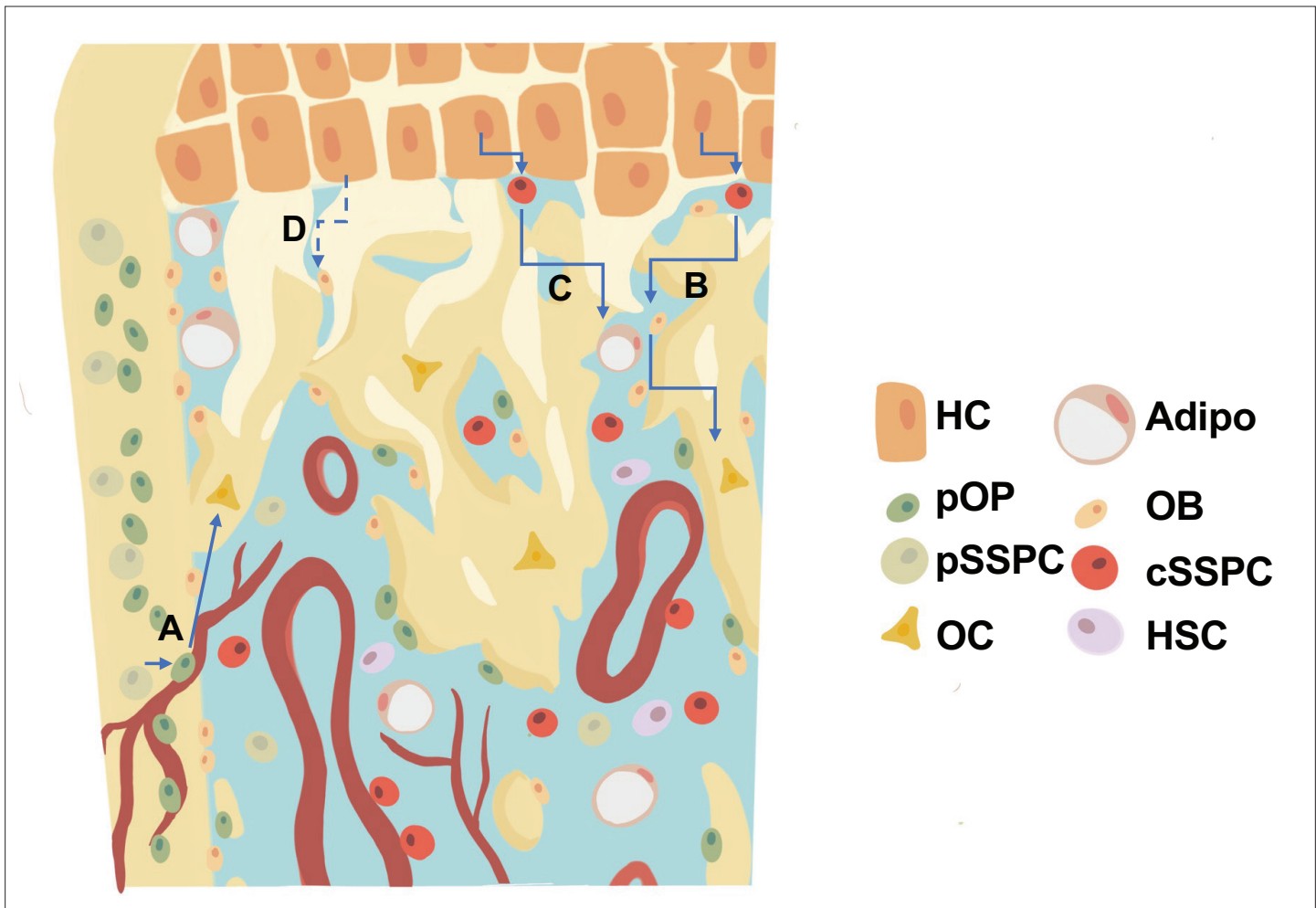

**Figure 9.** Revised model for the generation of trabecular/endocortical bone associated osteoblasts/osteocytes from multiple sources. (**a**) Perichondrial or periosteal SSPCs/osteoprogenitors (pSSPC/pOP) migrate into the marrow utilizing blood vessels and possess the ability to further differentiate into osteoblasts/osteocytes (OB/OC). (**b–c**) Hypertrophic chondrocytes (HC) dedifferentiate into chondrocyte derived SSPCs (cSSPC) with the capacity to generate osteoblasts/osteocytes (**b**) and adipocytes (Adipo) (**c**). (**d**) Data presented does not rule out the potential for the transdifferentiation of hypertrophic chondrocytes directly into osteoblasts/osteocytes. HSC = hematopoietic stem cells.

tracing and/or RNA-seq analyses or have utilized flow cytometric-based selection methods with antibodies against PDGFRα, CD200, and ITGAV among others followed by differentiation assays (*Boulais et al., 2018*; *Chan et al., 2015*; *Farahani and Xaymardan, 2015*; *Gulati et al., 2018*; *Mizuhashi et al., 2019*; *Ono et al., 2014*; *Worthley et al., 2015*; *Zhou et al., 2014a*). Generally, these studies have identified the presence of non-hematopoietic, non-endothelial multipotent mesenchymal cell populations with at least the capacity for osteogenic and/or adipogenic differentiation (*Robey et al., 2014*). It has also largely been assumed that these marrow-associated SSPCs are derived from the perichondrial/periosteal cells during skeletal development. Our study demonstrates that hypertrophic chondrocytes of the growth plate serve as a reservoir and alternative source of marrow-associated SSPCs likely utilized for rapid bone growth during skeletal development (*Figure 9*).

Specifically, our analyses of *Col10a1CreERT2; Rosa26*$^{fs-tdTomato}$ and *Col10a1Cre; Rosa26*$^{fs-tdTomato}$ skeletal rudiments at E16.5 identified not only hypertrophic chondrocytes (*Col10a1* expressing cells) and their descendant osteoblasts (*Bglap* expressing cells), but also hypertrophic chondrocyte derived SSPCs expressing *Pdgfra, Ly6a, Lepr*, and *Cxcl12*. A population of lineage negative, CXCL12-, PDGFRα+, SCA1+ SSPCs (PαS cells) have been identified as a quiescent population associated with vessels near the endosteum (*Morikawa et al., 2009*). These PαS cells are also capable of forming CFU-Fs, can differentiate into osteoblasts, chondrocytes, and adipocytes, while also being important for the migration and maintenance of HSCs (*Morikawa et al., 2009*; *Nusspaumer et al., 2017*). The appearance of these cells at E16.5 (but not 2 months) suggests that these might be either a more transient population derived from hypertrophic chondrocytes or are of an initial finite number and contribute early in development and are lost with age, as opposed to hypertrophic chondrocyte derived CXCL12-abundant reticular (CAR) cells (*Cxcl12*$^+$; *Ly6a1*$^-$) that continue to be present in our data at 2 months of age.

Our analyses of 2-month *Col10a1Cre; Rosa26*$^{fs-tdTomato}$ bone marrow identified not only hypertrophic chondrocyte-derived osteoprogenitors (*Sp7* expressing) (*Park et al., 2015*) and osteoblast populations, but also hypertrophic chondrocyte-derived marrow and vessel-associated cells expressing *Lepr, Pdgfra, Pdgfrb, Grem1, Cxcl12*, with several expressing *Itgav, Cd200, and Eng* (CD105) while lacking the expression of *Col10a1, Cre*, and most osteogenic genes. Since *Runx2, Pparg,* and *Adipoq* are also expressed throughout many of these cells at 2 months of age, we postulate that these hypertrophic chondrocyte-derived cells encompass populations of SSPCs with the capability of osteogenic and adipogenic differentiation. This indeed was confirmed via our in vitro and in vivo differentiation assays. Prior work has demonstrated that *Runx2* expressing and LEPR$^+$ progenitors are primed for osteoblastic differentiation (*Yang et al., 2017*), while LEPR$^+$ progenitors expressing *Adipoq* and *Pparg* are associated with adipogenic differentiation (*Zhong et al., 2020*). Priming of the SSPCs may allow for their rapid differentiation when in the proper supportive osteogenic/adipogenic niche or outside a niche that supports SSPC maintenance.

Since SSPC populations, including CAR cells, were thought to originate from perichondrial/periosteal cell sources and since LEPR+ cells constitute ~100% of CFU-Fs, we attempted to compare SSPCs/CAR cells derived from hypertrophic chondrocytes to those of other origins by utilizing *Col10a1Cre; Rosa26*$^{fs-tdTomato}$ bone marrow cells plated at clonal density (*Zhou et al., 2014a*). Nearly 50% of the CFU-Fs were originally derived from hypertrophic chondrocytes (tdTOMATO+), while the other half of CFU-Fs (tdTOMATO-) were likely derived from a perichondrial/periosteal origin. Additionally, when compared to SSPCs derived from other cell sources, the hypertrophic chondrocyte derived SSPCs were much more conducive to the formation of a bone ossicle with adipocytes and marrow upon transplantation, highlighting the supportive nature of these cells to vascular invasion and hematopoietic cell populations. Collectively, these data indicate for first time that growth plate hypertrophic chondrocytes may serve as a reservoir for the generation of SSPCs (including PαS cells) and osteoblasts during early embryonic development, but then primarily shift to CAR cell populations as the mice proceed through postnatal development and homeostasis.

Assessments of 2 M bulk RNA-seq data and EdU incorporation assays from CFU-F cultures, as well as cell cycle analysis of our scRNA-seq data, indicate an enhanced proliferative capacity for hypertrophic chondrocyte derived CFU-Fs and SSPCs/CAR cells. This data may be reflective of the fact that Yang et al. also identified the presence of both hypertrophic chondrocyte-derived *Col1a1*$^+$ *and Col1a1*$^-$ marrow-associated cells that were BrdU$^-$ proliferating cells, indicating that hypertrophic chondrocytes eventually re-enter the cell cycle during this cell fate transition (*Park et al., 2015*; *Yang*

*et al., 2014a*). Interestingly, our scRNA-seq dataset identified similar proportions of cycling cells in hypertrophic chondrocyte derived SSPCs/CAR cells and osteoblasts. This is in stark contrast to the more quiescent phenotype observed in most primitive SSPC populations that have been highlighted across numerous studies (*Tikhonova et al., 2019*; *Zhong et al., 2020*). However, recent work utilizing similar scRNA-seq approaches assessed proliferation rates of chondrocyte and perichondrial-derived SSPCs and osteoblast populations in the *Col2a1Cre; Rosa26*fs-tdTomato bone marrow and also indicated similar proliferative capacities between later committed progenitors and osteoblasts, which was greater than the most primitive of SSPCs (*Zhong et al., 2020*). Zhong et al., also utilized the *Adipo-qCre*ERT2; *Rosa26*fs-tdTomato model to investigate colony forming and proliferative potential of broad populations of *Adipoq*-expressing cells and descendants. These tdTOMATO+ cells contributed very little to CFU-Fs and exhibited minimal proliferation in vitro (*Zhong et al., 2020*). Tikhonova et al. also utilized scRNA-seq and identified a subset of cells from the *LeprCre; Rosa26*fs-tdTomato lineage that cluster into a high cycling group (*Tikhonova et al., 2019*), which may represent a more proliferative LEPR+ SSPC population derived from the cartilage growth plate. Finally, it is worth noting again that the in vivo contribution of *AdipoqCre; Rosa26*fs-tdTomato and *LeprCre; Rosa26*fs-tdTomato lineages to osteo-blasts and/or osteocytes is something that occurs over developmental time postnatally, while the *Col10a1Cre; Rosa26*fs-tdTomato lineage contributes to the osteoblast and osteocyte fates rapidly; as early as the embryonic period of skeletal development. This may also be attributed to the rapid or transient nature by which hypertrophic chondrocytes transit through the SSPC phase during these periods, such that the *LeprCre* or *AdipoqCre* do not have sufficient time or levels of *Cre* expression to induce recombination within this population. Also note that we do not observe *Adipoq* expression in the transitional cells at E16.5. Therefore, *LeprCre* and *AdipoqCre* may only be efficient at recombining floxed alleles within the less transient SSPC populations postnatally that continue to be maintained as SSPCs and subsequently slowly/weakly contribute to the osteogenic and adipogenic lineages during late postnatal/adult skeletal development and homeostasis.

While the data presented here demonstrates hypertrophic chondrocyte contribution to SSPCs, we note the importance of further studies to elucidate the mechanisms driving this process. Recent publications have suggested roles for pathways like WNT/β-catenin in balancing osteogenesis and adipogenesis from hypertrophic chondrocytes (*Tan et al., 2020*). Additionally, Sox9 was shown to be important for maintaining chondrocytes before changing fates toward the osteoblast lineage (*Haseeb et al., 2021*). Future studies will need to take into account the potential different signaling pathways that drive the dedifferentiation of hypertrophic chondrocytes to a SSPC fate versus the re-differen-tiation of hypertrophic chondrocyte derived SSPCs toward the osteogenic and adipogenic lineages. Additional studies have also demonstrated a potential transient embryonic and postnatal contribution of hypertrophic chondrocytes to developmental bone formation, since genetic deletion of *Runx2* within hypertrophic chondrocytes (and their descendants) initially reduces osteoblastic bone forma-tion that ultimately fully recovers around 3–6 weeks of age (*Qin et al., 2020*). While it is possible that the late postnatal formation is 'recovered' from osteoblasts derived purely from an alternate source (such as the periosteum), it is also possible that removal of *Runx2* within the hypertrophic chondro-cyte lineages induces some hypertrophic chondrocyte cell death, alters angiogenic/osteogenic factors (VEGFa among others) secreted from hypertrophic chondrocytes, and/or delays the re-differentiation of hypertrophic chondrocyte-derived SSPCs toward the osteoblast lineage; thus simply delaying post-natal bone formation by reducing the contribution of hypertrophic chondrocytes and their descen-dants to the earliest stages of bone development. Further studies are needed to more thoroughly understand these lineage contributions over time, as we demonstrate with the *Col10a1CreERT2; Rosa26*fs-tdTomato mouse line that there is a continued contribution of osteoblastic cells from hypertro-phic chondrocytes throughout embryonic development to at least approximately 4–5 months of age or skeletal maturity.

## Materials and methods
### Mouse strains

The *Col10a1Cre* mouse line was a generous gift from Dr. Klaus von der Mark and has been previously described (*Park et al., 2015*). The *Col10a1CreERT2* mouse line was a generous gift from Dr. Kathryn Cheah (University of Hong Kong). The *Rosa26*fs-tdTomato, *Pdgfra*H2B-GFP, *LeprCre*, and *AdipoqCre* mouse

lines were obtained from Jackson Laboratories (007909, 007669, 008320, and 028020, respectively). All animal work was approved by Duke University Institutional Animal Care and Use Committees (IACUC). Both sexes of mice were utilized for experiments.

## Tissue analysis

Harvest and sectioning: Animals were harvested at embryonic day 16.5 and postnatal day 0, 2–4 months, and 13 months of age. Tissues were washed in 1 x PBS and then fixed overnight with 4% paraformaldehyde (PFA - Sigma). P0 animals were treated overnight with 14% EDTA (Sigma), while 2 M old animals were treated for 10 days. Animals used for PERILIPIN labeling were not decalcified. Tissues were then transferred to 30% sucrose (Sigma) for 2 days. Samples were imbedded in Shandon Cryomatrix (ThermoFisher Scientific) and cryosectioned at 10 um, and slides were stored at –20 °C until staining.

For ENDOMUCIN and LEPR staining: Samples were prepared for staining as published (***Kusumbe et al., 2015***) 2 M animals were harvested and fixed overnight in 4% PFA. Samples were decalcified overnight in 14% EDTA. Tissues were transferred to 20% sucrose and 2% polyvinylpyrrolidone (PVP - Sigma) overnight. Samples were embedded in 8% gelatin (Sigma), 2% PVP, and 20% sucrose and cryosectioned at 30–200 um with slides stored at –20°C until staining.

Staining: Slides were brought to room temperature and rehydrated in PBS. Antigen retrieval was performed as shown in Appendix 1. Blocking was performed in 1% goat serum or 3% BSA. Slides were incubated with antibodies at concentrations in Appendix 1 overnight at 4 °C in a humidified dark box. Slides were washed and incubated with secondary antibodies (Invitrogen catalog number in Appendix 1 1:200) for 30–60 min at room temperature in the dark. Slides were imaged on a Leica DMI3000B microscope using a Leica DFC3000G camera with the latest Leica imaging suite, a Zeiss 710 or Zeiss 780 inverted confocal microscope running the latest ZEN software. Images were reconstructed and pseudocolored on Fiji and quantified using built-in quantification measures.

## Single-cell RNA sequencing

Cell isolation E16.5: Single cells were isolated by removing soft tissue on mechanical dissociation using scissors and then placed in 15 mL conical tubes with 2 mg/mL collagenase type II (Gibco 17101–015) in aMEM +20% FBS rotating at 37 °C. A pipet was used for further mechanical dissociation at 1- and 2 hr. Cells were removed at 2 hrs and subjected to red blood cell lysis for 30 s. Cells were stained with DAPI for 30 min on ice. Cells were washed with PBS and placed in PBS + 2% FBS for fluorescent activated cell sorting (FACS) on a B-C Astrios cell sorter. Gating was performed for single cells, DAPI, and tdTOMATO as shown in ***Figure 2—figure supplement 1a*** (Figure generated using FlowJo version 10). Cells were sorted into PBS + 10% FBS.

Cell isolation 2M-old: Single cells were isolated from bisected tibia and femur of 2 M *Col10a1Cre; Rosa26^{fs-tdTomato}* mice centrifuged at 4 °C twice pulsed up to 5000 x g. A 27 G needle attached to a syringe was then used to flush through the centrifuged bones. Bones and marrow were placed in 3 mg/mL of Collagenase D (Roche 11088866001) for 30 min in hypoxia. Red blood cells were lysed and strained through a 70 µm filter. Cells were stained with DAPI for 30 min on ice. Cells were washed with PBS and placed in PBS + 2% FBS for fFACS on a B-C Astrios cell sorter. Gating was performed for single cells, DAPI, and tdTOMATO as shown in ***Figure 2—figure supplement 1b*** (Figure generated using FlowJo version 10). Cells were sorted into PBS + 10% FBS.

10 x Transcriptome library prep: Cells were transferred to the Duke Molecular Physiology Institute Molecular Genomics core where they were combined with a master mix that contained reverse transcription reagents. The gel beads carrying the Illumina TruSeq Read one sequencing primer, a 16 bp 10 x barcode, a 12 bp unique molecular identifier (UMI) and a poly-dT primer were loaded onto the chip, together with oil for the emulsion reaction. The Chromium Controller partitions the cells into nanoliter-scale gel beads in emulsion (GEMS) within which reverse-transcription occurs. All cDNAs within a GEM, that is from one cell, share a common barcode. After the RT reaction, the GEMs were broken and the full length cDNAs cleaned with both Silane Dynabeads and SPRI beads. After purification, the cDNAs were assayed on an Agilent 4,200 TapeStation High Sensitivity D5000 ScreenTape for qualitative and quantitative analysis.

Enzymatic fragmentation and size selection were used to optimize the cDNA amplicon size. Illumina P5 and P7 sequences, a sample index, and TruSeq read two primer sequence were added via

End Repair, A-tailing, Adaptor Ligation, and PCR. The final libraries contained P5 and P7 primers used in Illumina bridge amplification. Sequence was generated using paired end sequencing (one end to generate cell specific, barcoded sequence and the other to generate sequence of the expressed poly-A tailed mRNA) on an Illumina NextSeq500 with 400 M read per run.

Analysis: The primary analytical pipeline for the SC analysis followed the recommended protocols from 10 X Genomics. Briefly, we demultiplexed raw base call (BCL) files generated by Illumina sequencers into FASTQ files, upon which alignment to the mouse reference transcriptome, filtering, barcode counting, and UMI counting were performed using the most current version of 10 X's Cell Ranger software. We used the Chromium cell barcode to generate feature-barcode matrices encompassing all cells captured in each library.

E16.5 timepoint: We sequenced a median of 4,868 genes per cell, a mean of 107,918 reads per cell, and reached a sequencing saturation of 44.6%. The secondary statistical analysis was performed using the latest R package of Seurat 4 (*Hao et al., 2020*). In Seurat, data was first normalized and scaled after basic filtering for minimum gene and cell observance frequency cut-offs ( > 200 genes, 15% mitochondrial genes, genes expressed in >5 cells). We thresholded *tdTomato* expression at two or higher to exclude a minimal number of cells that were separated via FACS but were *tdTomato* negative by gene expression. We observed some separation of clusters based on cell cycle and regressed out the cell cycle score during default SCTransform (*Hafemeister and Satija, 2019*).

After quality control procedures were complete, we performed linear dimensional reduction calculating principal components using the most variably expressed genes in our dataset (2000 variable genes, npcs = 35). Significant principal components for downstream analyses are determined through methods mirroring those implemented by Macosko et al, and these principal components were carried forward for two main purposes: to perform cell clustering and to enhance visualization (*Macosko et al., 2015*). Cells were grouped into an optimal number of clusters for de novo cell type discovery using Seurat's FindNeighbors and FindClusters functions (resolution = 0.5) graph-based clustering approach with visualization of cells being achieved through the use of UMAP, which reduces the information captured in the selected significant principal components to two dimensions. Differential expression of relevant genes was visualized on UMAP plots to reveal specific individual cell types. Additional downstream analyses included examining the cellular distribution and differential gene expression of a priori genes of interest and closer examination of genes associated within the identified 8 cell clusters. Monocle analysis was performed using the Monocle3 package after cds was generated from Seurat using SeuratWrappers (*Cao et al., 2019*; *Qiu et al., 2017a*; *Qiu et al., 2017b*; *Satija et al., 2020*; *Trapnell et al., 2014*). GEO accession number GSE190616.

Two-month timepoint: We sequenced a median of 2,820 genes per cell, a mean of 261,899 reads per cell, and a sequencing saturation of 92.3%. The secondary statistical analysis was performed using the last R package of Seurat 3 (*Butler et al., 2018*; *Konopka, 2019*; *R Development Core Team, 2013*; *Stuart et al., 2019*; *Wickham et al., 2020*). In Seurat, data was first normalized and scaled after basic filtering for minimum gene and cell observance frequency cut-offs (200–4000 genes, 7% mitochondrial genes, genes expressed in >5 cells). We then closely examined the data and performed further filtering based on a range of metrics in an attempt to identify and exclude possible multiplets (i.e. instances where more than one cell was present and sequenced in a single emulsified gel bead). We thresholded *tdTomato* expression at two or higher to exclude a minimal number of cells that were separated via FACS but were *tdTomato* negative by gene expression.

After quality control procedures were complete, we performed linear dimensional reduction calculating principal components using the most variably expressed genes in our dataset (2000 variable genes, npcs = 25). The genes underlying the resulting principal components are examined in order to confirm they are not enriched in genes involved in cell division or other standard cellular processes (subsetting out percent.mito). Significant principal components for downstream analyses are determined through methods mirroring those implemented by Macosko et al, and these principal components were carried forward for two main purposes: to perform cell clustering and to enhance visualization (*Macosko et al., 2015*). Cells were grouped into an optimal number of clusters for de novo cell type discovery using Seurat's FindNeighbors and FindClusters functions (resolution = 0.4) graph-based clustering approach with visualization of cells being achieved through the use of UMAP, which reduces the information captured in the selected significant principal components to two dimensions. Differential expression of relevant genes was visualized on UMAP plots to reveal specific

individual cell types. Additional downstream analyses included examining the cellular distribution and differential gene expression of a priori genes of interest and closer examination of genes associated within the identified 5 cell clusters. Cell cycle analysis of the scRNA-seq dataset was preformed using a gene list and commands packaged in Seurat (*Tirosh et al., 2016*; *Wickham, 2019*). Monocle analysis was performed using the Monocle3 package (num_dim = 25) (*Cao et al., 2019*; *Qiu et al., 2017a*; *Qiu et al., 2017b*; *Trapnell et al., 2014*). GEO accession number GSE179148.

## Colony-forming unit assays

The Colony Forming Unit – Fibroblastic (CFU-F) assay was performed as published, with the exception of placing cells in hypoxia based on a recent publication demonstrating reduction of contaminating cells (*Guo et al., 2020*; *Robey et al., 2014*). In short, cells were isolated from bisected tibia and femur of 2-month-old *Col10a1Cre; R26-tdTomato* mice centrifuged at 4°C twice pulsed up to 5 K xG. Following red blood cell lysis (Roche), cells were plated at 750 K cells per 15 cm tissue culture plate in complete media (a-MEM, 20% FBS, 1% penicillin-streptomycin) and glutamax (Gibco) after straining through a 70 µm filter. After incubation in hypoxia (2% oxygen, 5% $CO_2$, 37 C) for 3 hr, plates were washed three times with Hanks Balanced Salt Solution (Gibco) and replaced with complete media and glutamax. Media was changed on days 3 and 6. Colonies were utilized for analysis using a Leica DMI3000B microscope using a Leica DFC3000G camera with the latest Leica imaging suite, bulk RNA sequencing, or EdU labeling. TOMATO fluorescence was used to identify positive and negative colonies.

For CFU-OB, osteogenic differentiation was initiated after day 10 colony growth by replacing media with aMEM supplemented 10% FBS, 1% penicillin-streptomycin, 50 µg/mL L-ascorbic acid (Sigma-A4544), and 10 mM B-glycerophosphate (Sigma-G9422). Alkaline phosphatase and Von Kossa (Ricca) staining were performed per general histological procedures. For CFU-AD, adipogenic diffeentiation was performed as published; replacing media with aMEM supplemented with 10% FBS, 1% penicillin-streptomycin, 1.7 uM Insulin (Sigma), 1 µM Dexamethasone (Sigma), and 0.5 mM IBMX (Sigma) at day 10 colony growth (*Yu et al., 2019*). Oil red O (Sigma) staining was performed per general histological procedures.

## Bulk RNA sequencing

Colonies were imaged using a Leica DMI3000B microscope using a Leica DFC3000G camera with the latest Leica imaging suite to identify TOMATO positive and negative colonies grown as in the CFU-F assay section to day 10. Colonies were prepared using Corning 10 mm cloning cylinders and vacuum grease to provide single colony isolation. RLT buffer was pipetted directly into the cloning cylinder and then transferred to ethanol for processing using the RNeasy Micro kit (QIAGEN). RNA was provided to the Duke Center for Genomic and Computational Biology core for quality assessment and sequenced 50 bp single-end read on an Illumina HiSeq4000.

RNA-seq data was processed using the TrimGalore (https://github.com/FelixKrueger/TrimGalore) toolkit which employs Cutadapt to trim low-quality bases and Illumina sequencing adapters from the 3' end of the reads (*Martin, 2011*). Only reads that were 20nt or longer after trimming were kept for further analysis. Reads were mapped to the GRCm38v73 version of the mouse genome and transcriptome using the STAR RNA-seq alignment tool (*Dobin et al., 2013*; *Kersey et al., 2012*). Reads were kept for subsequent analysis if they mapped to a single genomic location. Gene counts were compiled using the HTSeq tool (*Anders et al., 2015*). Only genes that had at least 10 reads in any given library were used in subsequent analysis. Normalization and differential gene expression was carried out using the DESeq2 Bioconductor package with the R statistical programming environment (*Huber et al., 2015*; *Love et al., 2014*; *R Development Core Team, 2013*). The false discovery rate was calculated to control for multiple hypothesis testing. Biological thresholds were determined by taking the log2 expression value plotted on a histogram. A bimodal distribution is observed for low expressed genes followed by a larger Guassian distribution. The split point between these was used as a value cutoff for biological relevance. GEO accession number GSE179174.

## Gene ontology analysis

Gene ontology analysis was performed using geneontology.org Analysis Type: PANTHER Overrepresentation test released 02-24-2021. Annotation version and release date: release date 01-01-2021

and version 10.5281/zenodo.4437524 for biological processes using *Mus musculus* as the reference. Bulk RNA sequencing data was organized into log fold changes > 1 and p-value < 0.05 for input into the enrichment analysis. A negative log fold change in the dataset represents higher expression in TOMATO positive colonies while positive log fold change represents higher expression in TOMATO-negative colonies. Test type: Fisher's exact. Correction: Calculate False Discovery Rate (*Mi et al., 2019*; *Gene Ontology Consortium, 2021*; *Ashburner et al., 2000*).

### EdU incorporation assay

Colonies were grown as in CFU-F assays (in a six-well plate at 500 K cells per well instead of 15 cm plate) until day 5. EdU staining was performed following the protocol of Click-iT Plus EdU Cell Proliferation Kit (Invitrogen – C10637). In short, EdU was added to growth media to allow a half media change of the cells. The cells were incubated at in hypoxia at 37 °C for 4 hours. Cells were fixed, permeabilized, and treated with staining solution. Cells were imaged using a Zeiss Axio Observer Z1 microscope at ×5 magnification tiled across 70% of each well and stitched together using the latest Zen Blue software. Number of cells per colony calculations were performed at day 10 of plates grown in parallel to the EdU day 5 plate.

### Cell transplantations and ossicle formation assays

Cells from *Col10a1Cre; Rosa26$^{fs-tdTomato}$* mice at age 2 months were isolated as in Colony-Forming Unit Assays above. Cells were plated at high density on 10 cm plates for 6 days at 37 °C in hypoxia. Cells were trypsinized from the plates and were stained with DAPI for 30 min on ice. Cells were washed with PBS and placed in PBS + 2% FBS for FACS. Gating was performed for single cells, DAPI, and tdTOMATO similar to *Figure 2—figure supplement 1*. Cells that were tdTOMATO$^+$ and tdTOMATO$^-$ were isolated from each of 5 plates (each from an individual mouse) and transplanted into the kidney capsule (one transplant per mouse – two male/3 female) after being resuspended and absorbed into a Gelfoam matrix at 200–300 k cells per transplant. Transplants were isolated at 8 weeks post transplantation, washed in 1 x PBS, and then fixed overnight with 4% paraformaldehyde (PFA - Sigma). The samples were then decalcified overnight with 14% EDTA (Sigma). Tissues were then transferred to 30% sucrose (Sigma) for 2 days. Samples were imbedded in Shandon Cryomatrix (ThermoFisher Scientific) and cryosectioned at 20 μm, and slides were stored at –20 °C until staining. Staining was performed as above in Tissue Analysis.

### Flow cytometry

Following isolation techniques noted in the Colony Forming Unit Assay methods above, cells were resuspended in complete media and stained for 30–45 min in conjugated antibodies (noted in above table). Cells were spun down following staining and resuspended in PBS + 2% FBS. Flow cytometry was performed on a BD FACSCanto II machine.

### Statistical analysis

Statistical analyses were performed using two-tailed, unpaired Student's t-test. A p-value < 0.05 was considered significant.

## Acknowledgements

We acknowledge the assistance of the Duke Molecular Physiology Institute Molecular Genomics core for the generation of single cell RNA-seq libraries and the Duke Center for Genomic and Computational Biology for the bulk RNA-seq and sequencing of the scRNA-seq libraries. We also acknowledge Dr. Klaus von der Mark for generously providing the *Col10a1Cre* mouse line. Further, we acknowledge the Duke Light Microscopy Core Facility for their support with imaging and analysis, as well as, the Flow Cytometry Shared Resource Core at Duke for FACS and flow cytometric analysis. This work was supported in large part by R01 grants from the National Institutes of Health (NIH) and the National Institute of Arthritis and Musculoskeletal and Skin (NIAMS) Diseases to MJH (R01AR071722 and R01AR063071) and CMK (AR076325 and AR071967), as well as, by grants from the Research Grants Council and University Grants Council of Hong Kong: AoE/M-04/04, T12-708/12 N to KSEC.

## Additional information

### Competing interests

Kathryn Song Eng Cheah: Reviewing editor, *eLife*. The other authors declare that no competing interests exist.

### Funding

| Funder | Grant reference number | Author |
|---|---|---|
| NIH/NIAMS | R01AR071722 | Matthew J Hilton |
| NIH/NIAMS | R01AR063071 | Matthew J Hilton |

The funders had no role in study design, data collection and interpretation, or the decision to submit the work for publication.

### Author contributions

Jason T Long, Data curation, Formal analysis, Investigation, Writing - original draft, Writing - review and editing; Abigail Leinroth, Yihan Liao, Yinshi Ren, Tuyet Nguyen, Wendi Guo, Deepika Sharma, Data curation, Investigation, Writing - review and editing; Anthony J Mirando, Data curation, Investigation, Methodology, Writing - review and editing; Douglas Rouse, Data curation, Investigation; Colleen Wu, Courtney M Karner, Resources, Supervision; Kathryn Song Eng Cheah, Resources; Matthew J Hilton, Conceptualization, Funding acquisition, Project administration, Resources, Supervision, Writing - original draft, Writing - review and editing

### Author ORCIDs

Jason T Long (ID) http://orcid.org/0000-0002-6006-0932
Tuyet Nguyen (ID) http://orcid.org/0000-0002-8769-9955
Kathryn Song Eng Cheah (ID) http://orcid.org/0000-0003-0802-8799
Matthew J Hilton (ID) http://orcid.org/0000-0003-3165-267X

### Ethics

This study was performed in strict accordance with the recommendations in the Guide for the Care and Use of Laboratory Animals of the National Institutes of Health. All of the animals were handled according to and approved by the Duke University Institutional Animal Care and Use Committees (IACUC) (A068-20-03).

### Decision letter and Author response

Decision letter https://doi.org/10.7554/eLife.76932.sa1
Author response https://doi.org/10.7554/eLife.76932.sa2

## Additional files

### Supplementary files

• Transparent reporting form

• Source code 1. Code used for generation of Single Cell datasets and analysis at e16.5/2M of Col10a1Cre;Rosa26fs-tdTomato.

### Data availability

All raw data has been made available as source data files within the manuscript. All sequencing datasets are available via the Gene Expression Omnibus (GEO) under the accession numbers: GSE179174, GSE190616, and GSE179148.

The following datasets were generated:

| Author(s) | Year | Dataset title | Dataset URL | Database and Identifier |
|---|---|---|---|---|
| Long JT, Leinroth A, Liao Y, Ren Y, Mirando AJ, Nguyen T, Guo W, Sharma D, Rouse D, Wu C, Cheah KS, Karner CM, Hilton MJ | 2022 | Collagen10a1-Cre;Rosa26-tdTomato Bone marrow colony forming units | https://www.ncbi.nlm.nih.gov/geo/query/acc.cgi?acc=gse179174 | NCBI Gene Expression Omnibus, GSE179174 |
| Long JT, Leinroth A, Liao Y, Ren Y, Mirando AJ, Nguyen T, Guo W, Sharma D, Rouse D, Wu C, Cheah KS, Karner CM, Hilton MJ | 2022 | Collagen10-Cre;Rosa26-tdTomato e16.5 single cell RNA-sequencing | https://www.ncbi.nlm.nih.gov/geo/query/acc.cgi?acc=GSE190616 | NCBI Gene Expression Omnibus, GSE190616 |
| Long JT, Leinroth A, Liao Y, Ren Y, Mirando AJ, Nguyen T, Guo W, Sharma D, Rouse D, Wu C, Cheah KS, Karner CM, Hilton MJ | 2022 | Collagen10-Cre;Rosa26-tdTomato bone marrow single cell RNA-sequencing | https://www.ncbi.nlm.nih.gov/geo/query/acc.cgi?acc=GSE179148 | NCBI Gene Expression Omnibus, GSE179148 |

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

# Appendix 1

## Appendix 1—key resources table

| Reagent type (species) or resource | Designation | Source or reference | Identifiers | Additional information |
|---|---|---|---|---|
| Strain, strain background (*Mus musculus*) | *C57BL/6 J* | Jackson Laboratory | RRID:IMSR_JAX:000664 | |
| Genetic reagent (*Mus musculus*) | *Col10a1Cre* | DOI: 10.1242/bio.201411031 | | Dr. Klaus von der Mark |
| Genetic reagent (*Mus musculus*) | *Col10a1CreERT2* | This paper | | Dr. Kathryn Cheah |
| Genetic reagent (*Mus musculus*) | *Rosa26<sup>fs-tdTomato</sup>* | Jackson Laboratory | RRID:IMSR_JAX:007909 | |
| Genetic reagent (*Mus musculus*) | *Pdgfra<sup>H2B-GFP</sup>* | Jackson Laboratory | RRID:IMSR_JAX:007669 | |
| Genetic reagent (*Mus musculus*) | *LeprCre* | Jackson Laboratory | RRID:IMSR_JAX:008320 | |
| Genetic reagent (*Mus musculus*) | *AdipoqCre* | Jackson Laboratory | RRID:IMSR_JAX:028020 | |
| Sequence-based reagent | *Col10a1Cre/ ERT2_F* | DOI: 10.1242/bio.201411031 | | TTTAGAGCATTATT TCAAGGCA GTTTCCA Dr. Klaus von der Mark |
| Sequence-based reagent | *Col10a1Cre/ ERT2_R* | DOI: 10.1242/bio.201411031 | | AGGCAAATCTT GGTGTACGG Dr. Klaus von der Mark |
| Antibody | ENDOMUCIN (rat mAb) | Abcam | Cat# ab106100, RRID:AB_10859306 | IF (1:100), 0.3% Triton X-100 in 1 x PBS |
| Antibody | PERILIPIN (rabbit mAb) | Cell Signaling | Cat# 9349, RRID:AB_10829911 | IF (1:100) |
| Antibody | OSTEOCALCIN (rabbit pAb) | Millipore | Cat# AB10911, RRID:AB_1587337 | IF (1:200), 10 ug/mL Proteinase K |
| Antibody | OSTERIX (Rabbit pAb) | Abcam | Cat# ab22552, RRID:AB_2194492 | IF (1:400) |
| Antibody | CRE (Rabbit mAb) | Cell Signaling | Cat# 15036, RRID:AB_2798694 | IF (1:100), 0.3% Triton X-100 in 1 x PBS |
| Antibody | RFP (rabbit pAb) | Abcam | Cat# ab62341, RRID:AB_945213 | IF (1:100) |
| Antibody | LEPR (goat pAb) | R&D Systems | Cat# AF497, RRID:AB_2281270 | IF (1:50), 0.3% Triton X-100 in 1 x PBS |
| Antibody | PDGFRa (goat mAb) | R&D Systems | Cat# AF1062, RRID:AB_2236897 | IF (1:50), 0.3% Triton X-100 in 1 x PBS |
| Antibody | PDGFRb (rat mAb) | Invitrogen | Cat# 14-1402-81, RRID:AB_467492 | IF (1:50), 0.3% Triton X-100 in 1 x PBS |
| Antibody | Osteopontin (goat pAb) | R&D Systems | Cat# AF808, RRID:AB_2194992 | IF (1:100), 0.3% Triton X-100 in 1 x PBS |
| Antibody | PDGFRb (rat mAb) | Invitrogen | Cat# 17-1402-80, RRID:AB_1548752 | Flow Cytometry (1:25) |
| Antibody | CD45-FITC (rat mAb) | Biolegend | Cat# 103107, RRID:AB_312972 | Flow Cytometry (1:1000) |
| Antibody | CD45-APC (rat mAb) | Biolegend | Cat# 103111, RRID:AB_312976 | Flow Cytometry (1:1000) |
| Antibody | CD31-FITC (rat mAb) | Biolegend | Cat# 102405, RRID:AB_312900 | Flow Cytometry (1:1000) |
| Antibody | CD31-APC (rat mAb) | Biolegend | Cat# 102409, RRID:AB_312904 | Flow Cytometry (1:1000) |
| Antibody | TER119-FITC (rat mAb) | BD Biosciences | Cat# 561032, RRID:AB_10563083 | Flow Cytometry (1:500) |

*Appendix 1 Continued on next page*

*Appendix 1 Continued*

| Reagent type (species) or resource | Designation | Source or reference | Identifiers | Additional information |
|---|---|---|---|---|
| Antibody | TER119-APC (rat mAb) | Biolegend | Cat# 116211, RRID:AB_313712 | Flow Cytometry (1:250) |
| Antibody | DAPI (FLOW) | ThermoFisher | Cat# D1306, RRID:AB_2629482 | Flow Cytometry (1:1000) |
| Antibody | Goat anti-Mouse Alexa Fluor 488 pAb | Invitrogen | Cat# A-11001, RRID:AB_2534069 | IF (1:200) |
| Antibody | Donkey anti-Goat Alexa Fluor 488 pAb | Invitrogen | Cat# A-11055, RRID:AB_2534102 | IF (1:200) |
| Antibody | Goat anti-Rat Alexa Fluor 488 pAb | Invitrogen | Cat# A-11006, RRID:AB_2534074 | IF (1:200) |
| Antibody | Goat anti-Rabbit Alexa Fluor 594 pAb | Invitrogen | Cat# A-11037, RRID:AB_2534095 | IF (1:200) |
| Antibody | Goat anti-Rabbit Alexa Fluor 488 pAb | Invitrogen | Cat# A-11034, RRID:AB_2576217 | IF (1:200) |
| Antibody | Goat anti-Rat Alexa Fluor 647 pAb | Invitrogen | Cat# A-21247, RRID:AB_141778 | IF (1:200) |
| Antibody | Goat anti-Rabbit Alexa Fluor 488 pAb | Invitrogen | Cat# A-11008, RRID:AB_143165 | For *LepRCre;tdTomato* stained PERILIPIN only - IF (1:200) |
| Chemical compound, drug | Collagenase II | Gibco | 17101–015 | |
| Chemical compound, drug | Collagenase D | Roche | 11088866001 | |
| Chemical compound, drug | Polyvinylpyrrolidone | Sigma | P5288 | |
| Chemical compound, drug | Glutamax 100 x | Gibco | 35050–061 | |
| Chemical compound, drug | L-Ascorbic Acid | Sigma | A4544 | |
| Chemical compound, drug | B-glycerophosphate | Sigma | G9422 | |
| Chemical compound, drug | Insulin Human | Sigma | I3536 | |
| Chemical compound, drug | IBMX | Sigma | I7018 | |
| Chemical compound, drug | Dexamethasone | Sigma | D2915 | |
| Chemical compound, drug | Antigen Unmasking solution, Citrate Acid Based | Vector | H3300 | |
| Commercial assay, kit | Click-iT Plus EdU Cell Proliferation Kit | Invitrogen | C10637 | |
| Software, algorithm | FIJI | doi:10.1038/nmeth.2019 | Fiji, RRID:SCR_002285 | |
| Software, algorithm | Seurat | CRAN | Seurat, RRID:SCR_007322 | |
| Software, algorithm | Monocle3 | Github | Monocle3, RRID:SCR_018685 | |
| Software, algorithm | Trim Galore | Babraham Bioinformatics | Trim Galore, RRID:SCR_011847 | Felix Krueger |
| Software, algorithem | Cutadapt | Github | cutadapt, RRID:SCR_011841 | |
| Software, algorithm | STAR RNA-seq Alignment | Github | STAR, RRID:SCR_004463 | |
| Software, algorithm | HTSeq tool | Github | HTSeq, RRID:SCR_005514 | |

*Appendix 1 Continued on next page*

*Appendix 1 Continued*

| Reagent type (species) or resource | Designation | Source or reference | Identifiers | Additional information |
|---|---|---|---|---|
| Software, algorithm | DESeq2 | Bioconductor | DESeq2, RRID:SCR_015687 | |
| Software, algorithm | Gene Ontology | Geneontoloy.org | Gene Ontology, RRID:SCR_002811 | |
| Other | Silver Nitrate 10% | Ricca | 6830–4 | |
| Other | Oil Red O | Sigma | O0625 | |
| Other | Alizarin Red | Sigma | A5533 | |
| Other | Mayer's Hematoxylin solution | Electron Microscopy Sciences | 26043–06 | |
| Other | Eosin Y with Phloxine staining solution | Electron Microscopy Sciences | 26051–21 | |

