## [Editor Report]

The work reveals that a subpopulation of hypertrophic chondrocytes can form SSPCs that give rise to osteoblasts and adipocytes during skeletal development, providing new evidence to the current concept of chondrocyte-osteoprogenitors-osteoblast trans-differentiation.

---

## [Decision Letter]

**Decision letter after peer review:**

[Editors’ note: the authors submitted for reconsideration following the decision after peer review. What follows is the decision letter after the first round of review.]

Thank you for submitting the paper "Hypertrophic Chondrocytes Serve as a Reservoir for Unique Marrow Associated Skeletal Stem and Progenitor Cells, Osteoblasts, and Adipocytes During Skeletal Development" for consideration at *eLife*. Your initial submission has been reviewed by three peer reviewers, one of whom is a member of our Board of Reviewing Editors, and the evaluation has been overseen by a Senior Editor. Although the work is of interest, we regret to inform you that the findings at this stage are too preliminary for further consideration at *eLife*.

*Reviewer #1:*

Several prior studies showed that hypertrophic chondrocytes can give rise to osteoblast lineage cells during skeletal development and growth. In the present study, the authors, by conducted a characterization on the localization and lineage fate of the tdTomator+ cells using Col10a1CreERT2 ; R26-tdTomato lineage-tracing system, confirmed that hypertrophic chondrocytes gave rise to osteoblasts and other bone marrow associated cells. The authors then performed single cell RNA-sequencing on cells derived from 2-month-old Col10a1Cre; R26-tdTomato mice. They found that a subset of bone marrow tdTomato+ cells are skeletal stem and progenitor cells (SSPC) that contribute to osteogenic cells and the adipogenic lineage cells. Finally, the authors performed bulk RNA-sequencing on the tdTomoto+ and tdTomoto- CFU-Fs, and they found enriched expression of cell proliferation and vascular development genes within the tdTomoto+ CFU-Fs. These findings add new evidence to the current concept of chondrocyte-osteoprogenitors-osteoblast transdifferentiation and suggest hypertrophic chondrocytes may undergo dedifferentiate into SSPC. I have the following issues for the authors to address:

1. The lineage tracing study was conducted only in postnatal mice (P6) and chased for 2 weeks. Lineage tracing at different age groups (pre-natal or adult mice) would be useful. Moreover, more evidence is needed to claim that hypertrophic chondrocytes-derived cells undergo a process of dedifferentiation into SSPCs. For example, it is necessary to conduct fluorescence imaging to assess whether stem/progenitor markers (preferentially use another fluorescence reporter mouse strain) colocalize with tdTomato in bone/bone marrow tissue sections from Col10a1CreERT2 ; R26-tdTomato mice at different time points after tamoxifen treatment. Flow cytometry analyses of the expression of the stem/progenitor markers in sorted tdTomato+ cells would also be helpful.

2. Could the authors explain why the single cell RNA-seq were conducted using bone/bone marrow cells derived from 2-month-old non-inducible Col10a1Cre; R26-tdTomato mice? The Tamoxifen-inducible strain at early postnatal stage may make more sense.

3. In Figure 1l, 70-80% of OCN was tdTomato+ in Col10a1Cre; R26-tdTomato mice. However, in Figure 5 and Figure S10, tdTomoto+ and tdTomoto- colonies showed similar ability of osteogenesis. Please explain the discrepancy.

4. In Figure 5, the authors show that nearly 50% of CFU-F colonies derived from Col10a1Cre;R26-tdTomato mice. What is the age of the mice used? Is this age-associated?

5. In Figure S10, the authors assessed the in vitro osteogenic and adipogenic differentiation capacities of the colonies. The images of different staining results should be presented.

Recommendations for the authors:

1. I suggest that Figure S3, especially the staining data in Col10a1CreERT2;Rosa26-tdTomato mice, could be moved to the main text because it provides important evidence to support the dedifferentiation of the labeled tdTomato+ cells.

2. The mechanisms underlying the dedifferentiation of hypertrophic chondrocytes are interesting. This may be elaborated in the Discussion section.

*Reviewer #2:*

Strengths:

1. This study utilized a powerful combination of hypertrophic chondrocyte-specific lineage tracing models and single cell RNA-seq analyses, to uncover previously unknown identities of hypertrophic chondrocyte-derived bone marrow stromal cell populations.

2. This study took a closer look at Col10a1-creER-based lineage-tracing models to confirm that hypertrophic chondrocytes can transition into bone marrow stromal cells.

3. This study discovers heterogeneity of hypertrophic chondrocyte-derived cells in bone marrow, including osteoblasts and undifferentiated stromal cells that may overlap with skeletal stem and progenitor cells.

Weaknesses:

1. The major conclusion of this study on de-differentiation is mostly supported by single cell RNA-seq analyses of flushed and sorted bone marrow stromal cells, which does not capture the transition of hypertrophic chondrocytes.

2. The authors used Col10a1-CreER and Col10a1-Cre models interchangeably without much considering the important temporal factor. SSPCs that they defined in scRNA-seq experiments may not represent those generated by a recent transformation event. The transformation may have occurred at a much earlier development time point. Essentially all SSPCs in the endochondroal pathway are, after all, derived from chondrocytes. Therefore, these SSPCs may not represent a unique cell population.

3. The authors did not consider the heterogeneity of chondrocytes that are marked by Col10a1-Cre, which include hypertrophic chondrocytes at various locations of the growth plate.

4. This study did not directly show that hypertrophic chondrocyte-derived SSPC populations actually contribute to osteoblasts. These cells could be just bystanders, whereas direct transformation of hypertrophic chondrocytes to osteoblasts may be more significant.

5. The section pertaining to support of hematopoiesis was mostly irrelevant.

Recommendations for the authors:

1. Title:

It is unclear what the authors mean by "unique" marrow associated skeletal stem and progenitor cells. They defined these cells based on previously available markers as described in the main text; therefore, they are not "unique".

2. Abstract:

Generally speaking, this abstract is oversimplified and overstated, not capturing the essence of this study. The distinction between two models of lineage tracing (constitutive active vs. inducible) should be delineated.

Line 38: I am not sure which reference(s) the authors mean by saying that "SSPCs localized to the surrounding periosteum serve as the primary source of marrow associated SSPCs, osteoblasts, osteocytes, and adipocytes during skeletal development". The periosteum does not provide a precursor for marrow cell populations. The perichondrium is not a primary source of marrow cells.

Line 40: The authors should specify which genetic reporter mouse models they used in this study.

Line 42: What do the authors mean by this: "…unique marrow associated SSPCs, previously characterized as a CXCL12-abundant reticular (CAR) cells"? This sentence is contradictory. Moreover, the definition of CAR cells is well-established in the field.

3. Introduction:

Line 47-63: Olsen et al. 2000 is cited five times in the first paragraph. Although this is a valid book chapter to cite, they should identify the primary literature and also cite these papers accordingly.

Line 83-85: It is not true that the Acan-CreER model is more restricted to chondrocytes within the growth plate. It shows more recombination in the periosteum and other marrow components.

Line 109-110: It is not surprising that hypertrophic chondrocyte derived SSPCs are striking similar to marrow associated SSPCs, because, after all, most SSPCs are derived from chondrocytes in the endochondral pathway.

4. Results:

Line 128: IF staining in Figure 1d is not convincing.

Line 131-132: This point is not new and has been already shown in preceding studies. Although I appreciate their efforts to reproduce, these results are confirmatory and not entirely novel.

Line 153: Singe cell RNA-seq is not the right approach to reveal "multiple cell fates" of hypertrophic chondrocytes.

Line 160-161: To support this statement, the authors should include FACS plots in the paper.

Line 162: Why is the number of cells analyzed here so small (~1,000 cells)? This would limit the power of scRNA-seq analysis.

Line 217: Although interesting, the authors did not provide concrete evidence that de-differentiation generates "the most primitive SSCs". This is largely speculative.

Line 242-251: I am not sure how these discussions on support of hematopoiesis fit into the lines of discussion.

5. Discussion:

Line 390: I don't think that the authors showed here with genetic evidence that "hypertrophic chondrocytes undergo a process of dedifferentiation to form unique SSPCs populations". This piece of data would be critical to sustain the major conclusion of this study.

Line 403 (and the title): Why do the authors think that hypertrophic chondrocytes serve as a "reservoir and alternative source" of SSPCs? Their data clearly support the now well-accepted notion that these cells are a "primary" source of SSPCs.

Line 436: The authors are not the first group demonstrating that growth plate chondrocytes serve as a reservoir for CAR cells.

Line 466-469: The last conclusions appear to be vastly overstated.

*Reviewer #3:*

Authors examined the process of transdifferentiation of hypertrophic chondrocytes to osteoblasts or other marrow associated cells using the lineage tracing reporter mice (Col10a1 Cre-Rosa Tomato mice), which express Tomato in the hypertrophic chondrocytes and their descendants. They showed that more than 80% of bone marrow osteoblasts was derived from hypertrophic chondrocytes. They performed scRNA sequencing of Tomato-positive cells, and the gene expression profiles of most of the Tomato-positive cells are like those in skeletal stem and progenitor cells (SSPCs) and a niche cell for hematopoietic stem cells. It was confirmed by the staining of the Tamato-positive cells with immunohistochemistry using antibodies against PDGFRb, LEPR, and PDGFRa, which are expressed in SSPCs. Further, they showed that 25% of adipocytes and 37% of LepR+ cells were derived from Tamato-positive cells by immunohistochemistry. They also showed that half of the CFU-F was derived from Tamato-positive cells, and they had more proliferative activity than that derived from Tamato-negative cells. They concluded that hypertrophic chondrocytes undergo a process of dedifferentiation to generate SSPCs, which serve as a primary source of bone marrow osteoblasts and some source of adipocytes.

Strengths:

The scRNA sequencing of the transdifferentiated cells enabled the characterization of SSPCs derived from hypertrophic chondrocytes in detail.

Weaknesses:

1. They used immunohistochemistry for the quantitative analysis, including the frequencies of transdifferentiated osteoblasts and LepR+ cells. The immunohistochemistry using adult mouse samples is not easy because of the process of decalcification, which reduces antigen reactivity. They should use reporter mice driven by 2.3 kb Col1a1 promoter for the frequency of transdifferentiated osteoblasts, and LepR Cre Rosa mice for the frequency of LepR+ cells. Further, they prepared the samples by immersion fixation but not perfusion fixation. It is important to prepare the samples for immunohistochemistry by perfusion fixation to keep the antigen reactivity. They performed immunohistochemistry using OCN antibody to detect osteoblasts in Figure 1 and Supplementary Figure 8. The number of OCN-positive cells are too low, indicating that the immunohistochemical analysis is not working. Further, the pictures with higher magnification are required for the evaluation of the staining in Figure 1 and Supplementary Figures3, 6, 8, and the locations of the pictures with high magnification should be shown in the pictures with low magnification. In Supplementary Figure 3, most of the cells look Tomato single positive cells and PDGFRb, LEPR, or PDGFRa single positive cells. It is contradictory to Figure 3.

2. 37 % of LepR+ cells were derived from hypertrophic chondrocytes. However, more than 80% of OCN-positive cells were derived from hypertrophic chondrocytes. The discrepancy also seems to have been caused by the quantitative analysis using immunohistochemistry of LepR and OCN.

3. In Figure 2, the populations of osteocytes (cluster 1) and osteoblasts (cluster 2) were too small, indicating that the samples were not appropriately collected. It skews the understanding of dedifferentiation and transdifferentiation. It is recommended to perform scRNA sequencing at embryonic stage (E16.5-18.5) to confirm the appearance of dedifferentiated cells.

4. The frequency of osteoblasts derived from hypertrophic chondrocytes at 2 months of age was 83% in this paper, and it was described as consistent to the previous reports. However, it was 19-31% at 3 weeks of age in Yang et al. (Cell Res 24:1266, 2014), 62-63% at 1 month of age in Zhou et al. (PLoS Genet 10: e1004820, 2014), and 15% at 3 weeks of age in Qin et al. (PLoS Genet 16: e1009169, 2020). The frequencies were examined using reporter mice in the previous three papers. As the OCN immunostaining was not working, however, the frequency of this paper is not reliable. Even if it is reliable, the data is not consistent to the previous papers. Further, the last paper, which was not cited in this paper, also showed that bone volume becomes normal by 6 weeks of age in the absence of transdifferentiation of hypertrophic chondrocytes, indicating the transient requirement of the transdifferentiation at embryonic and neonatal stage. Referring the previous findings, precise discussion should be done.

5. The presence of the dedifferentiated cells does not mean that hypertrophic chondrocytes transdifferentiate into osteoblasts through the dedifferentiated cell stage. The data in this paper is consistent with the previous findings, which indicated that hypertrophic chondrocytes become osteoblasts, adipocytes, and stromal cells. Further, the differentiation of hypertrophic chondrocytes into Cxcl12-positive stromal cells was already shown in the previous paper (Nature 563: 254, 2018), although in detailed expression profiles of the stromal cells were provided in this paper.

6. In Figure 5G, are more than half of the cells in cluster 1 (osteocytes) S/G2/M phase?

Recommendations for the authors:

1. In line 131 on page 6, what are other osteogenic markers?

2. In line 147-148 on page 7, what does "loosely associated with the marrow" mean?

---

## [Author Response]

[Editors’ note: the authors resubmitted a revised version of the paper for consideration. What follows is the authors’ response to the first round of review.]

Reviewer #1:Several prior studies showed that hypertrophic chondrocytes can give rise to osteoblast lineage cells during skeletal development and growth. In the present study, the authors, by conducted a characterization on the localization and lineage fate of the tdTomator+ cells using Col10a1CreERT2 ; R26-tdTomato lineage-tracing system, confirmed that hypertrophic chondrocytes gave rise to osteoblasts and other bone marrow associated cells. The authors then performed single cell RNA-sequencing on cells derived from 2-month-old Col10a1Cre; R26-tdTomato mice. They found that a subset of bone marrow tdTomato+ cells are skeletal stem and progenitor cells (SSPC) that contribute to osteogenic cells and the adipogenic lineage cells. Finally, the authors performed bulk RNA-sequencing on the tdTomoto+ and tdTomoto- CFU-Fs, and they found enriched expression of cell proliferation and vascular development genes within the tdTomoto+ CFU-Fs. These findings add new evidence to the current concept of chondrocyte-osteoprogenitors-osteoblast transdifferentiation and suggest hypertrophic chondrocytes may undergo dedifferentiate into SSPC. I have the following issues for the authors to address:1. The lineage tracing study was conducted only in postnatal mice (P6) and chased for 2 weeks. Lineage tracing at different age groups (pre-natal or adult mice) would be useful. Moreover, more evidence is needed to claim that hypertrophic chondrocytes-derived cells undergo a process of dedifferentiation into SSPCs. For example, it is necessary to conduct fluorescence imaging to assess whether stem/progenitor markers (preferentially use another fluorescence reporter mouse strain) colocalize with tdTomato in bone/bone marrow tissue sections from Col10a1CreERT2 ; R26-tdTomato mice at different time points after tamoxifen treatment. Flow cytometry analyses of the expression of the stem/progenitor markers in sorted tdTomato+ cells would also be helpful.

To address these concerns, we have provided new inducible lineage tracing data at both embryonic (Figure 1a-c, 3g; E16.5) and postnatal/adult (Figure S1a-b; 2-months and 4-months) time points. In Figure 3g, we demonstrate that hypertrophic chondrocytes initially labeled at E13.5-15.5 transition into a marrow associated *Pdgfra*-expressing cells during bone development. These data were achieved by generating a *Col10a1Cre^ERT2^; R26-tdTomato* mouse line that incorporates a PDGFRa-H2B-GFP knock-in allele. This PDGFRa-H2B- GFP allele was also utilized with our *Col10a1Cre; R26-tdTomato* mouse line for flow cytometry of marrow associated cells at 2-months of age and demonstrated that greater than 70% of all of hypertrophic chondrocyte derived marrow associated cells or SSPCs are PDGFRa/GFP positive, which is consistent with our scRNA-seq data at this timepoint (Figure S4c)*.*

2. Could the authors explain why the single cell RNA-seq were conducted using bone/bone marrow cells derived from 2-month-old non-inducible Col10a1Cre; R26-tdTomato mice? The Tamoxifen-inducible strain at early postnatal stage may make more sense.

Due to the transient nature of the hypertrophic chondrocytes following single tamoxifen injections in *Col10a1Cre^ERT2^; R26-tdTomato* mice and their limited contribution to the marrow utilizing this strategy, we opted to use the *Col10a1Cre^ERT2^; R26-tdTomato* mouse line. Original inducible experiments did not yield sufficient cell numbers following tissue digestion, FACS, and downstream quality check/removals. We chose 2-months of age as a postnatal developmental time point in which the hypertrophic chondrocytes would have likely produced a potential variety of de-differentiated and or differentiated marrow associated cell types. We thought this would be ideal for capturing the full breadth of what hypertrophic chondroyctes were capable of becoming during skeletal development and homeostasis, in addition to the bone associated osteoblasts and osteocytes. To further address reviewers’ concerns, we have now performed an additional scRNA-seq experiment and analysis of E16.5 *Col10a1Cre; R26-tdTomato* long bones (cartilage, bone, and marrow tdTomato+ cells) and have confirmed prior 2-month scRNA-seq data that hypertrophic chondrocytes undergo a dedifferentiation process to generate SSPCs prior to becoming cells of the osteoblast lineage (Figure 2, 3, S2, and S3).

3. In Figure 1l, 70-80% of OCN was tdTomato+ in Col10a1Cre; R26-tdTomato mice. However, in Figure 5 and Figure S10, tdTomoto+ and tdTomoto- colonies showed similar ability of osteogenesis. Please explain the discrepancy.

The reviewer notes that 70-80% of OCN+ osteoblasts were tdTomato+, and that the tdTomato+ and tdTomato- colonies were roughly equal in numbers and osteogenic potential. So then why are most osteoblasts and osteocytes derived from hypertrophic chondrocytes? We attribute this to the rapid growth that is required during early skeletal development and the quick turnover of hypertrophic chondrocytes from the growth plate. This likely allows for faster contribution of the hypertrophic chondrocyte derived cells as compared to the invading periosteal associated osteo- progenitors.

4. In Figure 5, the authors show that nearly 50% of CFU-F colonies derived from Col10a1Cre;R26-tdTomato mice. What is the age of the mice used? Is this age-associated?

The 50% CFU ratio was performed at 2-months of age (the same timepoint as our adult scRNA-seq experiment). We have now performed an additional CFU analysis at 8-months of age (Figure 7) and observed relatively similar numbers.

5. In Figure S10, the authors assessed the in vitro osteogenic and adipogenic differentiation capacities of the colonies. The images of different staining results should be presented.

We performed Oil Red O and Von Kossa staining to quantify the in vitro differentiation capacity. These were first analyzed under a fluorescence microscope and marked on the plate for tdTomato positive and tdTomato negative. After the stain is performed that tdTomato signal is much weaker and in the case of Von Kossa is completely opaque because of the stain. In preparation for this re-submission we instead performed in vivo kidney capsule transplantation assays to truly assess their multilineage differentiation capacity. These in vivo data demonstrate that both SSPC populations are capable of forming osteoblasts that generate bone; however, only the hypertrophic chondrocyte derived SSPCs are capable of forming a complete vascularized bone organ with host derived marrow, and donor derived osteoblasts, osteocytes, some marrow associated cells, and adipocytes (Figure 8).

6. I suggest that Figure S3, especially the staining data in Col10a1CreERT2;Rosa26-tdTomato mice, could be moved to the main text because it provides important evidence to support the dedifferentiation of the labeled tdTomato+ cells.

We have now included new *Col10a1Cre^ERT2^; R26tdTomato* lineage tracing data in Figure 1 and 3. We have also now provided additional scRNA-seq data in Figure 2 and 3 that clearly identifies the transition of hypertrophic chondrocytes into a SSPC cell population prior to progressing to the osteoblast lineage. The additional supportive data previously supplied remains in Figure S3.

7. The mechanisms underlying the dedifferentiation of hypertrophic chondrocytes are interesting. This may be elaborated in the Discussion section.

A discussion of the potential mechanisms underlying the dedifferentiation of hypertrophic chondrocytes has been expanded within the Discussion section of the manuscript.

Reviewer #2:Strengths:1. This study utilized a powerful combination of hypertrophic chondrocyte-specific lineage tracing models and single cell RNA-seq analyses, to uncover previously unknown identities of hypertrophic chondrocyte-derived bone marrow stromal cell populations.2. This study took a closer look at Col10a1-creER-based lineage-tracing models to confirm that hypertrophic chondrocytes can transition into bone marrow stromal cells.3. This study discovers heterogeneity of hypertrophic chondrocyte-derived cells in bone marrow, including osteoblasts and undifferentiated stromal cells that may overlap with skeletal stem and progenitor cells.Weaknesses:1. The major conclusion of this study on de-differentiation is mostly supported by single cell RNA-seq analyses of flushed and sorted bone marrow stromal cells, which does not capture the transition of hypertrophic chondrocytes.

We have now performed E16.5 scRNA-sequencing that captures various populations of hypertrophic chondrocytes, as well as their transition through a SSPC stage prior to becoming osteoblast lineage cells (Figure 2 and 3). Imaging of long bone sections from E16.5 *Col10a1Cre^ERT2^; R26-tdTomatof/+; Pdgfra-H2B-GFP* mice (tamoxifen E13.5-15.5) have also demonstrated tdTomato/GFP double positive cells within the marrow space that were OCN-, indicating the presence of *Pdgfra* (SSPC marker) expressing cells that are derived from hypertrophic chondrocytes and are not osteoblasts (Figure 3g).

2. The authors used Col10a1-CreER and Col10a1-Cre models interchangeably without much considering the important temporal factor. SSPCs that they defined in scRNA-seq experiments may not represent those generated by a recent transformation event. The transformation may have occurred at a much earlier development time point. Essentially all SSPCs in the endochondroal pathway are, after all, derived from chondrocytes. Therefore, these SSPCs may not represent a unique cell population.

We disagree that all SSPCs are derived from chondrocytes.

SSPCs have been identified in the periosteum which originate only from common osteo-chondral or mesenchymal progenitors and not chondrocytes specifically during early skeletal development. We do; however, agree that there might be subclasses of chondrocytes that potentially contribute to SSPCs differently. A recent study by Mizuhashi, et al. (2019) found that borderline chondrocytes exist and rarely undergo hypertrophy or express *Col10a1*. Differing sources of SSPCs is further supported by our CFU analysis that indicate than nearly half of colonies or progenitors do not originate from hypertrophic chondrocytes and that hypertrophic chondrocyte derived SSPCs appear to have different multilineage capabilities when transplanted in vivo. With this in mind, we also performed scRNA-seq of tdTomato+ cells from E16.5 *Col10a1Cre; R26-tdTomato* long bones which captures the specific transitional period as hypertrophic chondrocytes give way for the formation of the nascent marrow cavity and endochondral bone, allowing us to demonstrate that some of the SSPCs are certainly derived from hypertrophic chondrocytes during skeletal development (Figure 2, 3, and S3). Some of the SSPCs identified at 2months of age may indeed be cells that were generated at earlier time-points and remain as progenitors within the marrow space, however new lineage tracing data at 2- and 4-months of age suggest that their contribution to marrow associated cells and the osteoblast lineage persists well into adulthood (Figure S1).

3. The authors did not consider the heterogeneity of chondrocytes that are marked by Col10a1-Cre, which include hypertrophic chondrocytes at various locations of the growth plate.

We believe the reviewer is asking if there might be regional differences between hypertrophic chondrocytes. We cannot exclude this possibility, but note that we see SSPCs derived from hypertrophic chondrocytes along the length of the metaphyseal/trabecular region, as well as, throughout the marrow cavity. The expression of the *Col10a1-*driven CRE is specific to cells only within the hypertrophic zone of the growth plate, and therefore demonstrate that the marrow derived SSPCs, osteoblast lineage cells, and adipocytes are derived from cells within this hypertrophic zone. We have now referenced this in our discussion.

4. This study did not directly show that hypertrophic chondrocyte-derived SSPC populations actually contribute to osteoblasts. These cells could be just bystanders, whereas direct transformation of hypertrophic chondrocytes to osteoblasts may be more significant.

While it is difficult to conclude the exact percentage of cells actively contributing to the osteoblasts from hypertrophic chondrocyte derived SSPCs, we observed in our E16.5 scRNA-seq analysis only a small population of *Bglap+* cells and trajectory analysis places them downstream of *Pdgfra, Lepr,* and *Cxcl12* expressing SSPCs (Figure 2 and 3). We also demonstrated that hypertrophic chondrocyte derived SSPCs are fully capable of generating marrow associated cells, osteoblasts, osteocytes, and adipocytes when transplanted under the kidney capsule (Figure 8). We have also added more detail to the Discussion section to acknowledge this potential bystander impact during normal skeletal development in vivo.

5. The section pertaining to support of hematopoiesis was mostly irrelevant.

This section has been removed.

While it is difficult to conclude the exact percentage of cells actively contributing to the osteoblasts from hypertrophic chondrocyte derived SSPCs, we observed in our E16.5 scRNA-seq analysis only a small population of *Bglap+* cells and trajectory analysis places them downstream of *Pdgfra, Lepr,* and *Cxcl12* expressing SSPCs (Figure 2 and 3). We also demonstrated that hypertrophic chondrocyte derived SSPCs are fully capable of generating marrow associated cells, osteoblasts, osteocytes, and adipocytes when transplanted under the kidney capsule (Figure 8). We have also added more detail to the Discussion section to acknowledge this potential bystander impact during normal skeletal development in vivo.

Recommendations for the authors:1. Title:It is unclear what the authors mean by "unique" marrow associated skeletal stem and progenitor cells. They defined these cells based on previously available markers as described in the main text; therefore, they are not "unique".

We have removed the word unique.

2. Abstract:Generally speaking, this abstract is oversimplified and overstated, not capturing the essence of this study. The distinction between two models of lineage tracing (constitutive active vs. inducible) should be delineated.

The abstract has been reworked to address this issue.

Line 38: I am not sure which reference(s) the authors mean by saying that "SSPCs localized to the surrounding periosteum serve as the primary source of marrow associated SSPCs, osteoblasts, osteocytes, and adipocytes during skeletal development". The periosteum does not provide a precursor for marrow cell populations. The perichondrium is not a primary source of marrow cells.

We disagree with the reviewer’s assessment that the periosteum does not provide precursors for marrow cell populations, since previous literature has shown that osteoblastic progenitors migrate into the marrow with blood vessels. Specific citations have been provided in the manuscript.

Line 40: The authors should specify which genetic reporter mouse models they used in this study.

Details regarding the genetic reporters have been added.

Line 42: What do the authors mean by this: "…unique marrow associated SSPCs, previously characterized as a CXCL12-abundant reticular (CAR) cells"? This sentence is contradictory. Moreover, the definition of CAR cells is well-established in the field.

We have fixed this contradiction.

3. Introduction:Line 47-63: Olsen et al. 2000 is cited five times in the first paragraph. Although this is a valid book chapter to cite, they should identify the primary literature and also cite these papers accordingly.

Citations have been added to cite primary literature of the original data.

Line 83-85: It is not true that the Acan-CreER model is more restricted to chondrocytes within the growth plate. It shows more recombination in the periosteum and other marrow components.

The original publication that documented the creation of the *AcanCre^ERT2^* mouse line (Henry, et al. Genesis 2009) demonstrates fairly restricted X-gal staining to chondrocytes when observed by a short chase during postnatal skeletal development.

Line 109-110: It is not surprising that hypertrophic chondrocyte derived SSPCs are striking similar to marrow associated SSPCs, because, after all, most SSPCs are derived from chondrocytes in the endochondral pathway.

We disagree with the reviewer’s assessment that the periosteum does not provide precursors for marrow cell populations, since prior literature has shown that osteoblastic progenitors migrate into the marrow with blood vessels. Further, our manuscript demonstrates that slightly more than half of the SSPCs are derived from the hypertrophic chondrocytes and interestingly our in vivo transplantation assays indicate important functional differences between these SSPC populations derived from various sources.

4. Results:Line 128: IF staining in Figure 1d is not convincing.

We have replaced this image with a more representative stained image.

Line 131-132: This point is not new and has been already shown in preceding studies. Although I appreciate their efforts to reproduce, these results are confirmatory and not entirely novel.

We understand that we are not providing novel findings with this replication of the data, but we wanted to replicate the data for context of the utility of the *Cre* lines and to draw the attention to the non-osteoblast associated cells. Additionally, the specific *Col10a1Cre^ERT2^* mouse line utilized in this study has not been previously published and we wanted to validate similar results with this model.

Line 153: Singe cell RNA-seq is not the right approach to reveal "multiple cell fates" of hypertrophic chondrocytes.

We utilize scRNA-seq as one method to confirm the multiple cell fates of hypertrophic chondrocytes, in addition to lineage tracing, immunostaining (IF), and flow cytometry.

Line 160-161: To support this statement, the authors should include FACS plots in the paper.

These have now been added to Figure S2.

Line 162: Why is the number of cells analyzed here so small (~1,000 cells)? This would limit the power of scRNA-seq analysis.

The number of cells analyzed was small because only approximately 1/2 of all SSPCs are derived from hypertrophic chondrocytes, which have been previously shown to be a very small population. Further with the smaller number we were able to increase the depth of sequencing to assess potential variation within the SSPC populations. The additional E16.5 scRNA-seq was performed with >2000 cells and provides further confirmation of this transition.

Line 217: Although interesting, the authors did not provide concrete evidence that de-differentiation generates "the most primitive SSCs". This is largely speculative.

This has been corrected.

Line 242-251: I am not sure how these discussions on support of hematopoiesis fit into the lines of discussion.

This section has been moved as we believe it is consistent with the transplantation data that revealed marrow formation ability of the hypertrophic derived SSPCs, consistent with previous literature of the importance of these hematopoietic regulatory factors.

5. Discussion:Line 390: I don't think that the authors showed here with genetic evidence that "hypertrophic chondrocytes undergo a process of dedifferentiation to form unique SSPCs populations". This piece of data would be critical to sustain the major conclusion of this study.

We have corrected (eliminated) the use of “unique” as the data is consistent with previously identified SSPCs, such as the CAR cells.

Line 403 (and the title): Why do the authors think that hypertrophic chondrocytes serve as a "reservoir and alternative source" of SSPCs? Their data clearly support the now well-accepted notion that these cells are a "primary" source of SSPCs.Line 436: The authors are not the first group demonstrating that growth plate chondrocytes serve as a reservoir for CAR cells.

We do not believe it to be a well-accepted notion that chondrocytes, specifically *Col10a1*-expressing hypertrophic chondrocytes give rise to SSPCs. Previous data to target chondrocytes (*Col2a1Cre* is expressed in cells outside of the growth plate and *AcanCre^ERT2^* can only provide a portion of the information as it is inducible). *Pthrp* labeled cells, appear to be a more progenitor population that can either give rise to columns of chondrocytes (closer to our model) or to borderline chondrocytes (largely *Col10a1*-negative cells). The columns of chondrocytes have been shown to give rise to some CXCL12+ cells; however, this data was limited in that it took months for contribution to a full column and then to CXCL12+ cells. Our data is the first to more completely demonstrate that *Col10a1*-expressing hypertrophic chondrocytes give rise to not only CXCL12+ cells but also Lepr+ cells, and PαS cells embryonically. Furthermore, we have provided a more robust transcriptomic signature of the SSPCs that are generated by hypertrophic chondrocytes at multiple developmental time points.

Line 466-469: The last conclusions appear to be vastly overstated.

We have reworked the conclusions of the manuscript based on the both the previous and new data.

Reviewer #3:Authors examined the process of transdifferentiation of hypertrophic chondrocytes to osteoblasts or other marrow associated cells using the lineage tracing reporter mice (Col10a1 Cre-Rosa Tomato mice), which express Tomato in the hypertrophic chondrocytes and their descendants. They showed that more than 80% of bone marrow osteoblasts was derived from hypertrophic chondrocytes. They performed scRNA sequencing of Tomato-positive cells, and the gene expression profiles of most of the Tomato-positive cells are like those in skeletal stem and progenitor cells (SSPCs) and a niche cell for hematopoietic stem cells. It was confirmed by the staining of the Tamato-positive cells with immunohistochemistry using antibodies against PDGFRb, LEPR, and PDGFRa, which are expressed in SSPCs. Further, they showed that 25% of adipocytes and 37% of LepR+ cells were derived from Tamato-positive cells by immunohistochemistry. They also showed that half of the CFU-F was derived from Tamato-positive cells, and they had more proliferative activity than that derived from Tamato-negative cells. They concluded that hypertrophic chondrocytes undergo a process of dedifferentiation to generate SSPCs, which serve as a primary source of bone marrow osteoblasts and some source of adipocytes.Strengths:The scRNA sequencing of the transdifferentiated cells enabled the characterization of SSPCs derived from hypertrophic chondrocytes in detail.Weaknesses:1. They used immunohistochemistry for the quantitative analysis, including the frequencies of transdifferentiated osteoblasts and LepR+ cells. The immunohistochemistry using adult mouse samples is not easy because of the process of decalcification, which reduces antigen reactivity. They should use reporter mice driven by 2.3 kb Col1a1 promoter for the frequency of transdifferentiated osteoblasts, and LepR Cre Rosa mice for the frequency of LepR+ cells. Further, they prepared the samples by immersion fixation but not perfusion fixation. It is important to prepare the samples for immunohistochemistry by perfusion fixation to keep the antigen reactivity. They performed immunohistochemistry using OCN antibody to detect osteoblasts in Figure 1 and Supplementary Figure 8. The number of OCN-positive cells are too low, indicating that the immunohistochemical analysis is not working. Further, the pictures with higher magnification are required for the evaluation of the staining in Figure 1 and Supplementary Figures3, 6, 8, and the locations of the pictures with high magnification should be shown in the pictures with low magnification. In Supplementary Figure 3, most of the cells look Tomato single positive cells and PDGFRb, LEPR, or PDGFRa single positive cells. It is contradictory to Figure 3.

We have performed flow analysis for PDGFRa utilizing another reporter line (PDGFRa-H2B-GFP) and antibody for PDGFRB and found a majority of tdTomato+ cells in the marrow are SSPCs (Figure S4c). We cannot fully understand the contribution to direct transdifferentiation and dedifferentiation with the currently available tools. We disagree that using a *Col1a1(2.3kb)* reporter would only demonstrate transdifferentiated but could also include osteoblasts that went through a SSPC intermediate. Additionally, a *LepRCre* combined with a genetic reporter would also not be usable as the CRE recombinase would affect the reporter already in the *Col10a1Cre;R26-tdTomato* line. Previous publications cited in this paper have utilized immersion fixation for immuno-staining and we disagree that perfusion fixation is absolutely required for preserving antigen reactivity. We have provided better images further demonstrating the co-labeling of cells.

2. 37 % of LepR+ cells were derived from hypertrophic chondrocytes. However, more than 80% of OCN-positive cells were derived from hypertrophic chondrocytes. The discrepancy also seems to have been caused by the quantitative analysis using immunohistochemistry of LepR and OCN.

The difference in rates of generating the LEPR+ (SSPCs) and OCN+ (osteoblasts) are likely independent processes. The rate at which LEPR+ populations differentiate, proliferate, and/or become quiescent may not be the same and therefore the contributions to the OCN+ osteoblasts could change over time. Additionally, we cannot rule out some transdifferentiation of hypertrophic chondrocytes which would increase the OCN+ population independent of LEPR+ population.

3. In Figure 2, the populations of osteocytes (cluster 1) and osteoblasts (cluster 2) were too small, indicating that the samples were not appropriately collected. It skews the understanding of dedifferentiation and transdifferentiation. It is recommended to perform scRNA sequencing at embryonic stage (E16.5-18.5) to confirm the appearance of dedifferentiated cells.

We purposefully performed only a short digest on the bones as to not obtain as many osteoblasts. We were interested more in the question of whether hypertrophic chondrocytes could dedifferentiate into an SSPC-type population. The reviewer is correct that we cannot draw a direct connection between the rate of dedifferentiation and transdifferentiation from this dataset. Additionally, we have performed E16.5 scRNA-sequencing, as noted by multiple reviewers.

4. The frequency of osteoblasts derived from hypertrophic chondrocytes at 2 months of age was 83% in this paper, and it was described as consistent to the previous reports. However, it was 19-31% at 3 weeks of age in Yang et al. (Cell Res 24:1266, 2014), 62-63% at 1 month of age in Zhou et al. (PLoS Genet 10: e1004820, 2014), and 15% at 3 weeks of age in Qin et al. (PLoS Genet 16: e1009169, 2020). The frequencies were examined using reporter mice in the previous three papers. As the OCN immunostaining was not working, however, the frequency of this paper is not reliable. Even if it is reliable, the data is not consistent to the previous papers. Further, the last paper, which was not cited in this paper, also showed that bone volume becomes normal by 6 weeks of age in the absence of transdifferentiation of hypertrophic chondrocytes, indicating the transient requirement of the transdifferentiation at embryonic and neonatal stage. Referring the previous findings, precise discussion should be done.

We have reworded the conclusions of this study in comparison to the previous studies noted by the reviewer. Yang, et al. utilized a different Cre line as well as used in situ hybridization to analyze overlap. Zhou, et al. utilized the same Cre as this study but looked at 1 month of age which could be different than our 2-month time point because of continued turnover of the hypertrophic cells and proliferation of osteoblasts although their numbers are consistent with our osteocyte quantifications. Qin, et al. provided numbers that are not comparable as the control for these quantifications have one allele of Runx2 deleted (controls were Col10a1Cre;R26- tdTomato;Runx2fl/+). The authors did not provide evidence that the heterozygous allele did not cause a phenotype of its own and also confounds the conclusion that transdifferentiation is only an embryonic/neonatal phenomenon.

5. The presence of the dedifferentiated cells does not mean that hypertrophic chondrocytes transdifferentiate into osteoblasts through the dedifferentiated cell stage. The data in this paper is consistent with the previous findings, which indicated that hypertrophic chondrocytes become osteoblasts, adipocytes, and stromal cells. Further, the differentiation of hypertrophic chondrocytes into Cxcl12-positive stromal cells was already shown in the previous paper (Nature 563: 254, 2018), although in detailed expression profiles of the stromal cells were provided in this paper.

See Reviewer 2 comment 4 for the transdifferentiation versus dedifferentiation conclusions. Additionally, we disagree that this paper demonstrated that hypertrophic chondrocytes were the source of CAR cells. The authors utilized a *Pthrp-Cre^ER^* model that targeted mSSC (based on their flow data) and no evidence of them being chondrocytes (only showing a much earlier state). The authors saw contribution to chondrocyte columns and what appear to be hypertrophic chondrocytes, but do not investigate this further. Additionally, their later paper 2019 JBMR showed that this *Pthrp-Cre^ER^* can label a borderline chondrocyte (*Col10a1*-negative) cell population associated with some contribution to CAR cells, albeit very minimally at an early postnatal time point. Furthermore, since this tool is not a constitutive CRE, the total contribution over time is very limited. When quantifications were performed the authors did not calculate the percentage of contribution, but rather simply noted that *Pthrp-Cre^ER^* derived cells are capable of generating CXCL12+ marrow associated cells. Finally, the *Pthrp-Cre^ER^* derived cells were very restricted in their potential as they could never give rise to adipocytes. We leave open the possibility that if these cells do contribute to hypertrophic chondrocytes later in development, they likely are not the only contributing cells as we have demonstrated that hypertrophic chondrocyte derived SSPCs can give rise to adipocytes.

6. In Figure 5G, are more than half of the cells in cluster 1 (osteocytes) S/G2/M phase?

We do observe some distribution across the cell cycle in cluster 1. Cluster 1 does not appear to be an osteocyte as osteoblasts can express the genes highlighted in Figure 4 and we do not observe *Sost* or other osteocyte associated genes in cluster 1, suggesting this is likely not an osteocyte population but rather a second osteoblast population, potentially on a differentiation path toward an osteocyte. Our digestion protocol was not designed to dissociate osteocytes, but rather focus on the marrow associated hypertrophic chondrocyte derived cell populations.

In line 131 on page 6, what are other osteogenic markers?

This has been reworded.

In line 147-148 on page 7, what does “loosely associated with the marrow” mean?

This has been reworded.